# Intratumoral SIRPα-deficient macrophages activate tumor antigen-specific cytotoxic T cells under radiotherapy

Zhen Bian [1,2,3], Lei Shi [1,2,3], Koby Kidder[2,3], Ke Zen[2], Charlie Garnett-Benson[2] & Yuan Liu[1,2 ✉]

Radiotherapy (RT)-induced tumoricidal immunity is severely limited when tumors are well-established. Here, we report that depleting SIRPα on intratumoral macrophages augments efficacy of RT to eliminate otherwise large, treatment-resistant colorectal (MC38) and pancreatic (Pan02 and KPC) tumors, inducing complete abscopal remission and long-lasting humoral and cellular immunity that prevent recurrence. SIRPα[-deficient] macrophages activated by irradiated tumor-released DAMPs exhibit robust efficacy and orchestrate an anti-tumor response that controls late-stage tumors. Upon RT-mediated activation, intratumoral SIRPα[-deficient] macrophages acquire potent proinflammatory features and conduct immunogenic antigen presentation that confer a tumoricidal microenvironment highly infiltrated by tumor-specific cytotoxic T cells, NK cells and inflammatory neutrophils, but with limited immunosuppressive regulatory T cells, myeloid derived suppressor cells and post-radiation wound-healing. The results demonstrate that SIRPα is a master regulator underlying tumor resistance to RT and provide proof-of-principle for SIRPα[-deficient] macrophage-based therapies to treat a broad spectrum of cancers, including those at advanced stages with low immunogenicity and metastases.

[1] Center of Diagnostics and Therapeutics, Georgia State University, Atlanta, GA, USA. [2] Program of Cancer and Immunology, Georgia State University, Atlanta, GA, USA. [3] These authors contributed equally: Zhen Bian, Lei Shi, Koby Kidder. ✉email: yliu@gsu.edu

Radiation therapy (RT) has long been cemented as a pillar of cancer therapy especially against solid tumors[1,2]. In the best-case scenario, RT not only eliminates irradiated tumors but also induces an abscopal response that, through RT-induced anti-tumor cytotoxic CD8 T cell (Tc) immunity, systemically clears nonirradiated distal metastases[3,4]. Unfortunately, this curative response to RT, though reported occasionally, does not commonly occur in clinical settings[5]. Most RT modalities, even those combined with immunomodulatory agents (e.g., checkpoint inhibitors) to augment antitumor immunity, fall short of inducing adequate immune-mediated tumoricidal activity, especially when tumors are large, poorly immunogenic, and/or harbor a strong immunosuppressive milieu[6–9]. To improve clinical outcomes, additional understanding of mechanisms by which tumors resist RT, as well as innovative methods to enhance RT-induced immune responses, are needed.

Signal-regulatory protein α (SIRPα) is a myeloid leukocyte-expressed inhibitory regulator whose canonical function is to inhibit professional phagocytes (mainly macrophages) from phagocytosing self-cells including tumor cells via interacting with the self-recognition marker CD47[10]. Despite the fact that many cancers exploit this mechanism through increasing CD47 expression, and thereby evade attack by phagocytes, we previously found that mere depletion of the CD47-SIRPα axis or SIRPα-mediated inhibitory signaling does not lead to phagocytosis[11]. In line with this finding, established tumors even at advanced stages expressing no or very low CD47 broadly exist across different cancer types in patients (https://www.proteinatlas.org/ENSG00000196776-CD47/pathology). However, when Sirpα$^{-/-}$ macrophages were activated by certain proinflammatory cytokines or TLR agonizts, they displayed a strong capacity to rapidly phagocytize self-cells regardless of CD47 expression[11]. Given that macrophages are abundant in MC38 and PDA tumors[12] and irradiation (IR)-induced damage-associated molecular patterns (DAMPs), such as HMGB1 and calreticulin (CRT), are able to activate phagocytes via TLR or LRP-1 signaling pathways[13–15], we postulated that IR-induced DAMPs in the tumor microenvironment (TME) could stimulate Sirpα$^{-/-}$ macrophages to phagocytose tumor cells and thereby control tumor growth in Sirpα$^{-/-}$ mice.

In this preclinical study exploring RT combination strategies, we assess local tumor RT against well-established colorectal adenocarcinoma MC38 and pancreatic ductal adenocarcinoma (PDA) Pan02 and KPC in Sirpα$^{-/-}$ mice. The results reveal that RT-activated Sirpα$^{-deficient}$ phagocytes function less as enhanced phagocytes toward tumor cells, but more so as proinflammatory initiators and adept antigen-presenting cells (APC), which conduct immunogenic antigen presentation in situ to robustly activate tumor-specific memory T cells and Tc cytotoxicity. This anamnestic anti-tumor response, coupled with the activated Sirpα$^{-deficient}$ macrophage-mediated proinflammatory response, transforms the TME into a potent tumoricidal niche highly infiltrated by tumor-specific Tc, NK cells, and tissue-damaging neutrophils, while diminishing regulatory T cells (Tregs) myeloid-derived suppressor cells (MDSCs) and other immunosuppressive mechanisms.

## Results

**Local RT achieves curative responses in Sirpα$^{-/-}$ mice with late-stage tumors.** Our previous study demonstrated that Sirpα depletion strongly enhanced the capacity of macrophages to phagocytize self-cells regardless of CD47 expression[11]. In line with this, when activated, bone marrow-derived macrophage (BMDMs) obtained from Sirpα$^{-/-}$ mice displayed a strong capacity to phagocytize CD47-expressing MC38 cancer cells, whereas BMDMs from WT mice failed to engulf cancer cells

under the same treatment (Supplementary Fig. 1a). These stimuli activating macrophages included LPS, IL-1β, IL-6, CpG, and HMGB1. CpG oligodeoxynucleotide was used to mimic irradiated tumor-released DNA fragment and HMGB1 was a DAMP released by damaged cancer cells[16–18]. Interestingly, the addition of anti-CD47 Ab or Sirpα-ex fusion protein enabled WT BMDMs to phagocytize cancer cells, but to a significantly lesser degree than Sirpα$^{-/-}$ BMDMs under the same stimulation. Given that IR-damaged cancer cells often produce DAMPs, which can activate macrophage inflammatory responses[19], we thus compared Sirpα$^{-/-}$ and WT BMDM phagocytosis toward MC38 cells with or without IR treatment. In contrast to WT BMDMs, which did not phagocytize MC38 cells with or without 8 Gy treatment, Sirpα$^{-/-}$ BMDMs displayed great phagocytic capacity toward IR-treated MC38 but not non-treated MC38 cells (Supplementary Fig. 1b). Anti-CD47 Ab or Sirpα-ex fusion protein also promoted phagocytosis of irradiated MC38 cells by WT BMDMs but to a lesser extent than that of Sirpα$^{-/-}$ BMDMs. Immunofluorescent staining showed that following IR, CRT expression significantly increased on the surface of MC38 cells (Supplementary Fig. 1c). As downregulation of cell surface levels of CD47 is associated with cell apoptosis[20], no reduction of CD47 on irradiated MC38 cells (Supplementary Fig. 1c) suggests that these irradiated cancer cells may be damaged but not apoptotic. We also assessed whether conditioned medium derived from irradiated MC38 cells affected macrophage phagocytosis toward cancer cells. After treatment with conditioned medium from irradiated MC38 cells, Sirpα$^{-/-}$ BMDMs exhibited significant phagocytosis toward MC38 cells, whereas the same treatment did not drive WT BMDMs to phagocytize cancer cells (Supplementary Fig. 1d). These results support the notion that IR induces cancer cells to release DAMPs that activate Sirpα$^{-/-}$ macrophages, but not WT macrophages, to phagocytize CD47-expressing cancer cells. In line with this, incubating Sirpα$^{-/-}$ macrophages with irradiated MC38 cells or their conditioned medium induced higher levels of phosphorylated P65 and P38, two downstream signaling molecules indicative of TLR pathway activation[14], than in WT macrophages (Supplementary Fig. 1e).

Given that Sirpα$^{-/-}$ macrophages were able to phagocytize cancer cells in vitro after activation by irradiated tumor-produced DAMPs, we next explored the role of Sirpα$^{-/-}$ macrophages combined with IR in mouse tumor clearance. Sirpα$^{-/-}$ and WT mice were subcutaneously (s.c.) engrafted with $5 \times 10^3$, $5 \times 10^4$, or $5 \times 10^5$ MC38 cells per mouse. When inoculated with a lower number of MC38 cells ($5 \times 10^3$ cells/mouse), Sirpα$^{-/-}$ mice exhibited significantly slower MC38 tumor growth compared to WT mice, suggesting that Sirpα$^{-/-}$ mice have inherently greater immune control against syngeneic MC38 in vivo (Fig. 1a). However, when the number of engrafted cells was increased ($5 \times 10^4$, $5 \times 10^5$, or $2 \times 10^6$ cells/mouse), MC38 tumors grew at a similar rate in Sirpα$^{-/-}$ and WT mice (Fig. 1a). These observations suggest that the phagocytic activity of Sirpα$^{-/-}$ macrophages towards MC38 cells during engraftment is likely abolished by a larger quantity of anti-inflammatory cytokines, such as IL-10[11], released by a greater number of cancer cells.

We, therefore, established syngeneic tumors in Sirpα$^{-/-}$ and WT mice by s.c. engrafting mice with $5 \times 10^5$ MC38 cells per mouse. As shown in Fig. 1b, syngeneic MC38 tumors were grown to different volumes (small: <100 mm$^3$; medium: 100-400 mm$^3$; large: 400–600 mm$^3$) and then a single fraction of non-ablative, local X-ray radiation (4, 8, or 15 Gy) was given at a dose rate of 1.2 Gy/min. In agreement with the previous report by Jones et al.[21], we found that RT dose-dependently suppressed MC38 tumor growth in WT mice when MC38 tumors were small (<100 mm$^3$). However, when tumor volume reached 150 mm$^3$ or above, even a high dose of RT failed to control tumors in WT mice, which continued growing and rapidly reached the humane

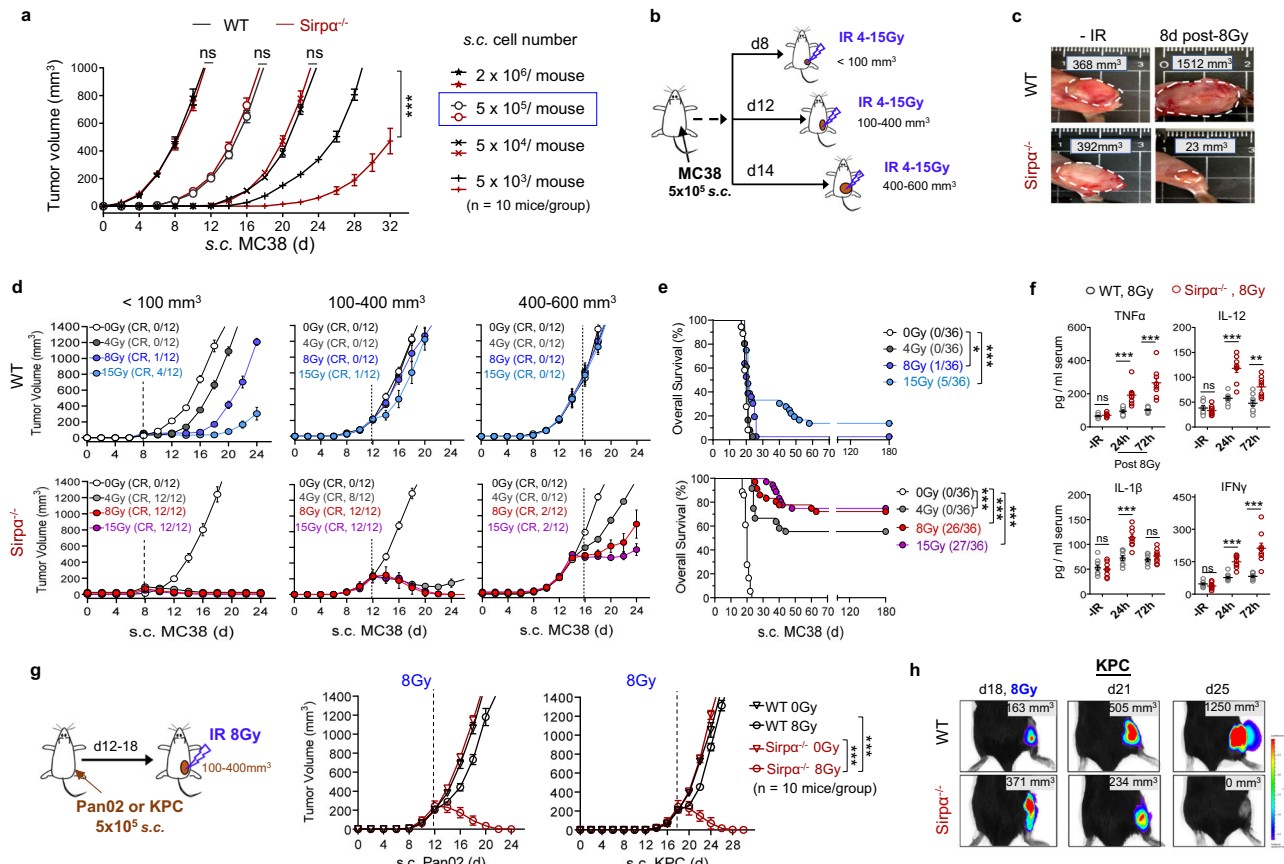

**Fig. 1 Local RT eliminates late-stage MC38 tumors in Sirpα$^{-/-}$ mice but not WT mice. a** Growth of MC38 tumor in WT and Sirpα$^{-/-}$ mice. Various numbers of MC38 cells were engrafted into WT and Sirpα$^{-/-}$ mice via *s.c.* injection. Data from three independent experiments (3–4 mice/group in each experiment) are presented as mean ± SEM. $n = 10$ mice in each group. ***$P < 0.001$. **b** RT scheme. MC38 cells ($5 \times 10^5$) were subcutaneously (*s.c.*) engrafted into the right flank of WT or Sirpα$^{-/-}$ mice and X-ray irradiation (IR) of various doses were administered when tumor volume was <100 mm$^3$, 100–400 mm$^3$, or 400–600 mm$^3$. **c** Representative images of MC38 in WT and Sirpα$^{-/-}$ mice before and after 1 × 8 Gy. Data represent three independent experiments (3 mice/group in each experiment). **d**, **e** Curves of MC38 tumor growth and mouse survival rate. A single fraction of IR was given when MC38 tumors were <100 mm$^3$, 100–400 mm$^3$, or 400–600 mm$^3$. Tumor volume (**d**) from three independent experiments (4 mice/group in each experiment) are presented as mean ± SEM. Mouse survival rate (**e**) was recorded up to 6 months post-IR and the data from **d** were pooled and grouped together by the RT dose. *$P = 0.0262$, ***$P < 0.001$. **f** Serum cytokines before and after RT. Data from three independent experiments (3 mice/group in each experiment) are presented as individual values and mean ± SEM. $n = 9$ mice in each group. ns not significant, **$P < 0.01$, ***$P < 0.001$. **g** Pan02 and KPC tumor growth in WT and Sirpα$^{-/-}$ mice. Pan02 or KPC cells were engrafted ($5 \times 10^5$, s.c.) into the right flank of WT or Sirpα$^{-/-}$ mice, and a single fraction of 8 Gy were given when tumors were 100–400 mm$^3$. Data from three independent experiments (3-4 mice/group) are presented as mean ± SEM. $n = 10$ mice in each group. ***, $P < 0.001$. **h** Representative images of luciferase-expressing KPC tumors in WT and Sirpα$^{-/-}$ mice before and after 1 × 8 Gy. Data represent three independent experiments (3 mice/group in each experiment). $P$ values were calculated by one-way ANOVA with Tukey's post hoc test (**a**, **f**, and **g**) or one-way ANOVA with log-rank (Mantel–Cox) test (**e**). Source data are provided as a Source Data file.

endpoint (Fig. 1c–e). These results were consistent with studies by others[22,23], indicating that established MC38 tumors highly resist RT. In stark contrast, a single fraction of RT (4, 8, or 15 Gy) conferred robust responses in Sirpα$^{-/-}$ mice. For small tumors (<100 mm$^3$), any of the three RT doses completely eliminated MC38 tumors. For medium tumors (100–400 mm$^3$), each RT regimen well-controlled tumor growth. For large tumors (400–600 mm$^3$), RT still induced significant dose-dependent inhibition of tumor growth. In most cases, cessation of tumor growth in Sirpα$^{-/-}$ mice occurred immediately after IR, followed by durable regression and complete clearance of tumors in 4–12 days (Fig. 1c, d). Indeed, all Sirpα$^{-/-}$ mice with small tumors, as well as most with medium tumors, that received local RT displayed completed responses, survived without apparent long-term adverse effects, and remained tumor-free for the rest of the study (>180 days) (Fig. 1e). In line with its role in inducing rapid tumor regression, high dose rate RT (2 Gy/min, 20 Gy)

exaggerated the inflammatory responses in Sirpα$^{-/-}$ mice, which sometimes compromised mouse survival. All tumor-irradiated Sirpα$^{-/-}$ mice developed inflammatory responses with an elevated serum level of cytokines TNFα, IL-1β, IL-12, and IFNγ, whereas WT mice treated with the same RT regimen did not exhibit similar increases in cytokines (Fig. 1f). In a similar fashion, we engrafted WT and Sirpα$^{-/-}$ mice with Pan02 or KPC ($5 \times 10^5$ cells/mouse), and then when their tumors grew to 100–400 mm$^3$, they were treated with a single dose of 8 Gy IR (Fig. 1g, left). The results showed that a single dose of 8 Gy IR completely eliminated tumors in Sirpα$^{-/-}$ mice but not in WT mice (Fig. 1g, right). We also engrafted mice with luciferase-expressing KPC cells, and as shown by whole-body imaging (Fig. 1h), tumor growth was totally controlled by a single dose of 8 Gy IR in Sirpα$^{-/-}$ mice but not in WT mice. These differences demonstrated the exceptional tumor responsiveness to RT in Sirpα$^{-/-}$ mice.

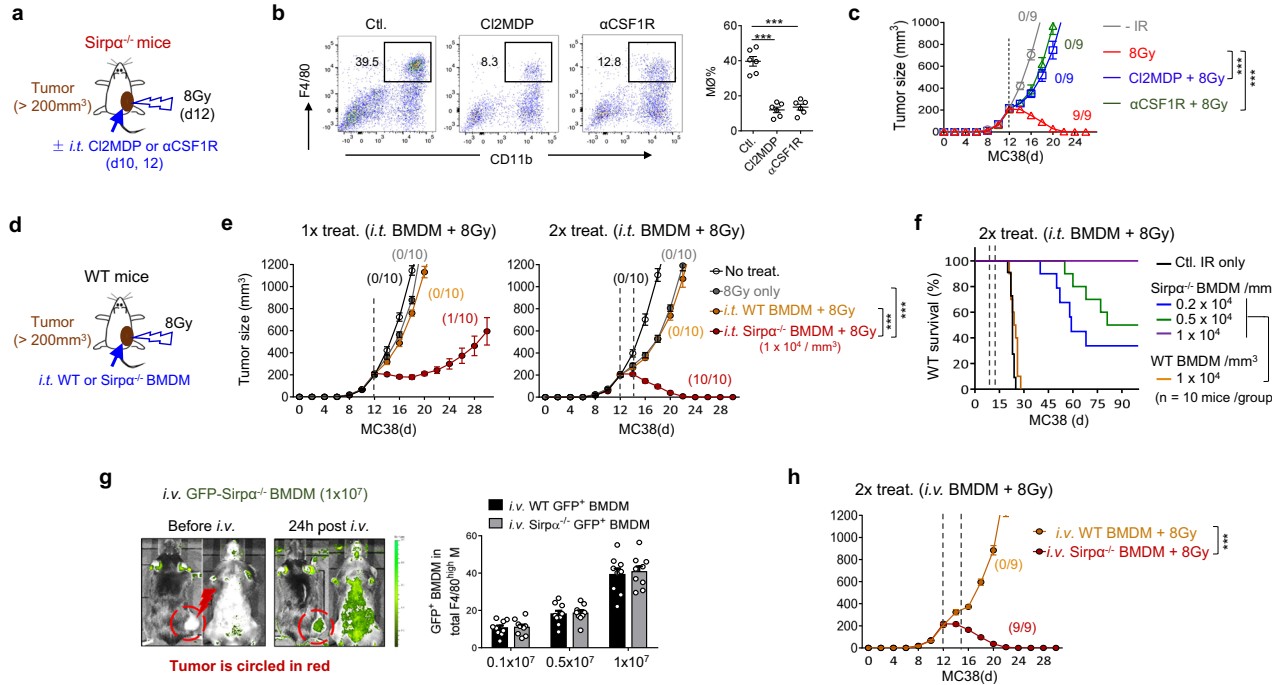

**Fig. 2 Sirpα$^{-/-}$ macrophages confer complete responses after IR. a–c** Depletion of intratumoral macrophages diminished RT efficacy in Sirpα$^{-/-}$ mice. MC38 tumor-bearing Sirpα$^{-/-}$ mice were administered Cl2MDA-liposomes (Cl2MDP) or anti-CSF receptor antibodies (αCSF1R) to deplete macrophages starting 2 days prior to IR (8 Gy) (**a**). Depletion of intratumoral macrophages by Cl2MDP or αCSF1R was assessed by flow cytometry (**b**). Data from two independent experiments (3 mice/group in each experiment) are presented as mean ± SEM. $n = 6$ mice in each group. Tumor growth (**c**) in Sirpα$^{-/-}$ mice was monitored after IR with or without macrophage depletion. Data from three independent experiments (3 mice/group in each experiment) are presented as mean ± SEM. $n = 9$ mice in each group. ***$P < 0.001$. **d, f** Combining adoptive infusion of Sirpα$^{-/-}$ BMDMs with RT conferred tumor elimination in WT mice. As depicted in the therapeutic scheme (**d**), MC38 tumors in WT mice were treated once (1×) or twice (2×, 3-days interval) with intratumorally (i.t.) injected Sirpα$^{-/-}$ or WT (Sirpα$^+$) BMDMs (1 × 10$^4$ per mm$^3$ tumor mass) followed by IR (8 Gy). Tumor size (**e**) and mouse survival rate (**f**) were analyzed. Data from three independent experiments (3–4 mice/group in each experiment) are presented as mean ± SEM. $n = 10$ mice in each group. ***$P < 0.001$. **g** Percentage of transferred Sirpα$^{-/-}$ BMDMs (GFP$^+$) in total intratumoral macrophages 12 h after i.v. injection of Sirpα$^{-/-}$ BMDMs (GFP$^+$) (0.1–1 × 10$^7$ per mouse). Data from three independent experiments (3 mice/group in each experiment) are presented as mean ± SEM. $n = 9$ mice in each group. **h** MC38 tumor growth in WT mice after 2× i.v. injection of BMDMs in combination with IR (8 Gy). Data from three independent experiments (3 mice/group) are presented as mean ± SEM. $n = 9$ mice in each group. ***$P < 0.001$. $P$ values were calculated by one-way ANOVA with Tukey's post hoc test (**b, c, e, g**) or one-way ANOVA with log-rank (Mantel–Cox) test (**f**). Source data are provided as a Source Data file.

**Intratumoral Sirpα$^{-/-}$ macrophages predicate the complete response to RT.** To determine whether intratumoral Sirpα$^{-/-}$ macrophages dictate the efficacy of RT in Sirpα$^{-/-}$ mice, we depleted intratumoral macrophages using Cl2MDA-liposomes or an antibody against the CSF1 receptor (αCSF1R)[24,25] prior to local IR (Fig. 2a). Flow cytometry analysis confirmed that both Cl2MDA-liposomes and αCSF1R significantly depleted intratumoral CD11b$^+$F4/80$^+$ macrophages (Fig. 2b). As shown in Fig. 2c, both Cl2MDA-liposomes and αCSF1R antibodies abrogated the efficacy of RT with respect to suppressing tumor growth in and enhancing the survival rate of Sirpα$^{-/-}$ mice, suggesting a key role for intratumoral Sirpα$^{-/-}$ macrophages in responding to RT. To further test this, we examined whether the adoptive transfer of Sirpα$^{-/-}$ macrophages into established MC38 tumors in WT mice could improve their responses to RT. Either intratumoral injection (i.t.) or intravenous administration (i.v.) of Sirpα$^{-/-}$ BMDMs were employed (Fig. 2d). The rationale for i.v. the administration was based on the fact that inactivated Sirpα$^{-/-}$ macrophages would not phagocytose self-cells in peripheral circulation[11] and that tumors could secrete chemokine CCL2 to recruit macrophages[26]. Both routes of administration led to intratumoral infiltration of Sirpα$^{-/-}$ BMDMs (Supplementary Fig. 2a), while neither method to infuse Sirpα$^{-/-}$ BMDMs caused adverse reactions such as anemia (Supplementary Fig. 2b). As shown in Fig. 2e, f, intratumoral infusion of Sirpα$^{-/-}$ BMDMs dose-dependently reversed tumor resistance to

RT in WT recipient mice. Two rounds of treatment comprising 8 Gy RT and i.t. infusion of Sirpα$^{-/-}$ BMDMs, which achieves an equal quantity of intratumoral Sirpα$^{-/-}$ BMDM and endogenous intratumoral WT macrophages (~1 × 10$^4$ cells per mm$^3$ of tumor volume), led to complete regression of MC38 tumors larger than 200 mm$^3$ in WT mice and 100% survival rates. For i.v. injection of Sirpα$^{-/-}$ BMDMs into tumor-bearing WT mice, we first monitored the recruitment of fluorescently labeled BMDMs (GFP$^+$) into engrafted MC38 tumors and found that when 1 × 10$^7$ BMDMs (GFP$^+$) per mouse were injected, they comprised nearly 40% of the total number of intratumoral macrophages (Fig. 2g). As shown in Fig. 2h, two rounds of treatment with i.v. injected Sirpα$^{-/-}$ BMDMs plus 8 Gy IR eliminated MC38 tumors. These results collectively demonstrate that the presence of Sirpα$^{-/-}$ macrophages in tumors predicates the enhanced response to RT.

As the cognate ligand for SIRPα, the "self-recognition" marker CD47 regulates phagocytosis of tumor cells by macrophages and CD47 blockade has been applied to treat various tumors[27,28]. Here we also examined the effect of CD47 blockade on RT-induced tumor elimination in WT mice using anti-CD47 antibody (αCD47; miap301) or murine SIRPα extracellular domain (mSIRPα.ex) fusion protein, which block the interaction between CD47 and SIRPα. As depicted in Supplementary Fig. 3, αCD47 and mSIRPα.ex were combined with RT to treat the MC38 tumors in WT mice when tumor sizes were >200 mm$^3$ or

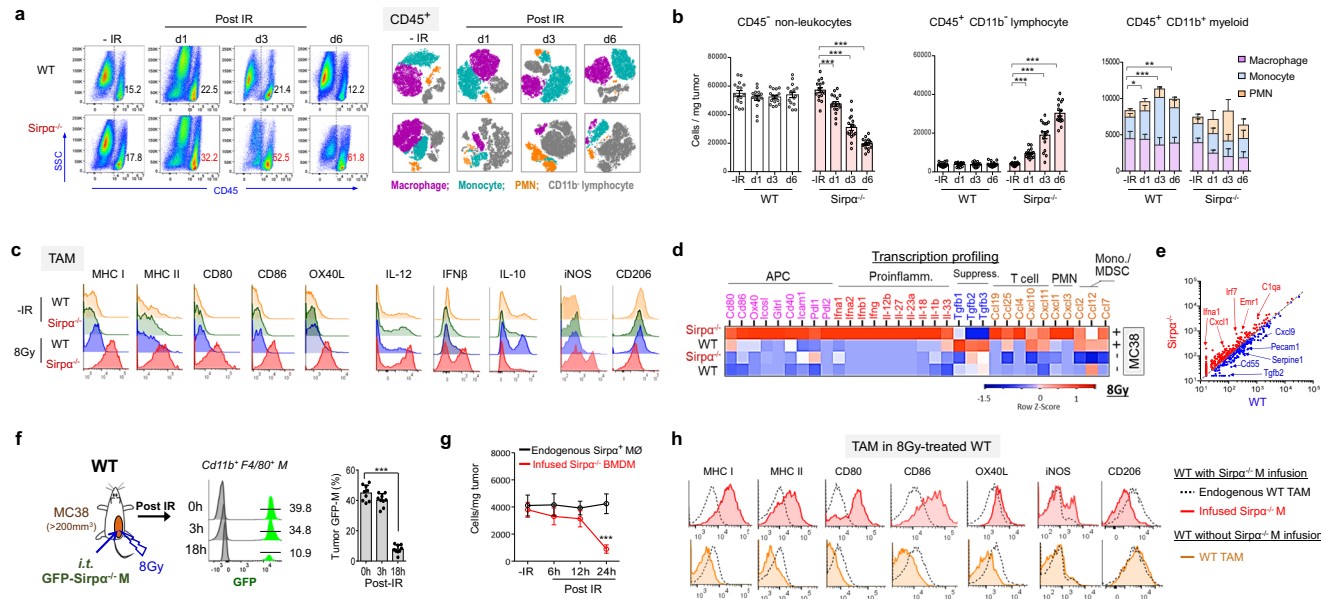

**Fig. 3 Irradiation-activated Sirpα$^{-/-}$ macrophages drive a proinflammatory TME. a** MC38 TME in (tumor > 200mm$^3$) WT and Sirpα$^{-/-}$ mice prior to and after 1 × 8 Gy were analyzed for CD45$^+$ leukocytes (left panel), in which other immune populations were displayed by t-SNE (right panel). Data are representative of five independent experiments ($n = 5$ mice/group in each experiment). **b** Number of CD45$^-$ non-leukocytes, CD45$^+$CD11b$^-$ lymphocytes, CD45$^+$ CD11b$^-$ myeloid cells, and different myeloid cells subpopulation (F4/80$^{high}$Ly6C$^-$ macrophage, F4/80$^+$Ly6C$^+$ monocyte and Ly6G$^+$ PMN) in WT and Sirpα$^{-/-}$ MC38 TME. Data from five independent experiments (3–4 mice in each experiment) are presented as mean ± SEM. $n = 16$ mice in each group. *$P = 0.023$, **$P = 0.008$, ***$P < 0.001$. **c** Analyses of tumor-associated macrophages (TAM) in MC38 tumors of WT and Sirpα$^{-/-}$ mice. TAM in MC38 tumors of WT and Sirpα$^{-/-}$ mice were analyzed for antigen presentation machinery, inflammatory phenotypes, and M1/M2 marker prior to (−IR) and 12 h post-IR (8 Gy). Cell-surface staining: MHC-I/II, CD80/86, OX40L, and CD206; intracellular staining: IL-12, IFNβ, IL-10, and iNOS. Data represent three independent experiments. $n = 10$ mice in each group. **d, e** Tumor mRNA profiling before and 12 h post-IR by Nanostring (nCounter Mouse Immunology Panel). Heatmap (**d**) and scatterplot (**e**) depict the differential expression of genes involved in antigen presentation and production of proinflammatory or anti-inflammatory cytokines/chemokines. $n = 4$ mice in each group. **f, g** GFP-positive Sirpα$^{-/-}$ BMDM (GFP-Sirpα$^{-/-}$ M) was infused (i.t., 1 × 10$^4$/mm$^3$) into MC38 tumors in WT recipients. The percentage of infused GFP-Sirpα$^{-/-}$ M within total TAM (**f**) and their number (**g**) were analyzed at different time points post-IR. ***$P < 0.001$. Data from three independent experiments (3–4 mice in each experiment) are presented as mean ± SEM. $n = 9$ mice (**f**) and 10 mice (**g**) in each group. **h** Analyses of intratumoral macrophages (MØ) in MC38 tumors of WT mice with or without Sirpα$^{-/-}$ BMDM infusion following 1 × 8 Gy. Data are representative of three independent experiments. $n = 10$ mice in each group. $P$ values were calculated one-way ANOVA with Tukey's post hoc test (**b, f**, and **g**). Source data are provided as a Source Data file.

<100 mm$^3$. To our surprise, the CD47 blockade did not recapitulate the compelling anti-tumor efficacy of RT conferred by Sirpα$^{-/-}$ BMDM infusion, despite both modalities disrupting the CD47-SIRPα axis. These reagents combined with RT only resulted in moderately delayed tumor progression but not durable regression. The limited efficacy of CD47-blockade when treating well-established (>200 mm$^3$) tumors aligns with observations by other studies, which reported that CD47-blockade alone or its combination with checkpoint inhibitors or RT largely achieve only partial responses against small (<100 mm$^3$) MC38 or other tumors[29–34]. The disparate efficacies of intratumoral Sirpα$^{-/-}$ macrophages vs. CD47-blockade imply that SIRPα-mediated regulation has a component independent of CD47 binding, which likely plays a key role in controlling tumor responses to RT.

**Activated Sirpα$^{-/-}$ macrophages following IR reshape the TME into a tumoricidal niche.** To explore the mechanism underlying the exceptional tumoricidal capacity of RT-activated intratumoral Sirpα$^{-/-}$ macrophages, we analyzed the TME in MC38-bearing WT and Sirpα$^{-/-}$ mice following IR. As shown in Fig. 3a, b, the post-IR TME in MC38-bearing Sirpα$^{-/-}$ mice were rapidly infiltrated by much greater numbers of leukocytes (CD45$^+$) compared to the TME in WT mice. By day 3 to day 6 post-IR, the infiltrating leukocytes in the Sirpα$^{-/-}$ TME, of which

the majority were lymphocytes, already outnumbered cancer cells, which rapidly reduced as tumor size shrank. Interestingly, cell subpopulation analysis of CD45$^+$ leukocytes showed that intratumoral Sirpα$^{-/-}$ macrophages rapidly disappeared after IR, occurring prior to detectable tumor regression, whereas endogenous WT (Sirpα$^+$) macrophages in WT mice did not reduce in number. The rapid reduction of Sirpα$^{-/-}$ macrophages in the TME following IR argues that the RT-induced tumor elimination in Sirpα$^{-/-}$ mice maybe not due to direct phagocytosis or clearance of tumor cells by Sirpα$^{-/-}$ macrophages. Notable changes in the post-IR Sirpα$^{-/-}$ TME also included diminishment of monocytes or MDSC but a significant increase in proinflammatory PMN. Flow cytometry analyses of intratumoral Sirpα$^{-/-}$ macrophages prior to their disappearance at 12 h post-IR found that Sirpα$^{-/-}$ macrophages manifested robust proinflammatory features and immunogenic antigen presentation machinery with increased cell surface MHC-I, MHC-II, CD80, CD86, and OX40L and intracellular IL-12 and IFNβ (Fig. 3c). Compared to WT macrophages, Sirpα$^{-/-}$ macrophages exhibited more of an M1 phenotype, with increased iNOS but decreased CD206 expression (Fig. 3c). We also analyzed BMDMs from Sirpα$^{-/-}$ and WT mice prior to tumor engraftment (Supplementary Fig. 4). Though there was no difference in phenotype prior to proinflammatory stimulation, activated Sirpα$^{-/-}$ BMDMs expressed higher levels of MHC-I, MHC-II, CD80,

CD86, OX40L, and iNOS but a lower level of CD206 than WT BMDMs.

The inflammatory profiles of bulk MC38 tumors in WT and Sirpα$^{-/-}$ mice with or without IR were further analyzed by Nanostring transcription profiling. As shown in Fig. 3d, e, nanostring transcription profiling revealed a strikingly altered Sirpα$^{-/-}$ TME shortly after IR, with wide-ranging increases in the transcription of proinflammatory cytokines (IFNα/β/γ, IL-1α/β, IL-12, IL-18, and IL-33), immunogenic antigen presentation co-stimulatory molecules (CD80, CD86, OX40, IcosL, GITRL, and CD40), T cell and neutrophil chemokines (CCL19/25/4 and CXCL1/3), as well as other notable molecules essential for tumor immunity (Mx1, IRF1, IRF7, etc.). Meanwhile, immunosuppressive cytokines such as TGFβ1/2/3 were substantially reduced in the Sirpα$^{-/-}$ TME after IR, signifying a phenotypic shift away from wound-healing toward proinflammation[9,35]. In contrast, irradiated tumors in WT mice showed prominent induction of TGFβ cytokines and monocyte chemokines CCL2 and CCL12 but weak proinflammatory transcription; their endogenous Sirpα$^{+}$ macrophages also lacked the capacity for immunogenic antigen presentation and exhibited increased IL-10 expression, suggesting they further strengthened immunosuppression. Nanostring transcription profiling of the TME was also performed in Sirpα$^{-/-}$ and WT mice engrafted with Pan02 or KPC tumors, and similar results were observed revealing wide-ranging increases in the transcription of proinflammatory cytokines, immunogenic antigen presentation co-stimulatory molecules, T cell and neutrophil chemokines (CCL19/25/4 and CXCL1/3), as well as other essential molecules for anti-tumor immunity (Supplementary Fig. 5). We next analyzed the TME in MC38-bearing WT mice following i.t. injection of GFP-Sirpα$^{-/-}$ macrophages (GFP Sirpα$^{-/-}$ M) (Fig. 3f). As shown, rapid reduction of GFP-Sirpα$^{-/-}$ macrophages but not endogenous Sirpα$^{+}$ macrophages in the TME following IR also occurred in WT mice. The time-course analyses during the 24 h window post-IR revealed that the intratumoral Sirpα$^{-/-}$ macrophage population remained unchanged until 12 h post-IR, but rapidly reduced thereafter (Fig. 3g). Furthermore, i.t. injection of Sirpα$^{-/-}$ macrophages led to an increase in the expression of immunogenic antigen presentation machinery (MHC-I/II, CD80/86, and OX40L) on endogenous WT tumor-associated macrophages (Fig. 3h). Taken together, these results demonstrate that IR-activated intratumoral Sirpα$^{-/-}$ macrophages likely execute their tumoricidal function through reshaping the TME into a proinflammatory tumoricidal niche.

**RT-activated Sirpα$^{-/-}$ macrophages robustly induce tumor-specific cytotoxic CD8 T cells.** Among the differences between tumor milieus comprising Sirpα$^{-/-}$ macrophages and those without Sirpα$^{-/-}$ macrophages, the robust expansion of tumor-infiltrating cytotoxic T cells (Tc) in Sirpα$^{-/-}$ TME after IR was the most striking. As shown in Fig. 4a, b, IR of MC38 tumors in Sirpα$^{-/-}$ mice led to the rapid expansion of intratumoral Tc. Flow cytometry further demonstrated that intratumoral Tc went from ~10% of the total intratumoral leukocytes to nearly 40% in 24 h and further increased to 50–70% by day 3–6 (Fig. 4a). Immunostaining of irradiated tumor tissues from Sirpα$^{-/-}$ mice (3-days post-IR) revealed numerous Tc distributing in the tumor core and throughout the invasive edge (Fig. 4b). Not only did these Tc expand to a large quantity, but they also displayed high levels of granzyme B (GranzB) expression, suggestive of potent cytotoxicity, and specificity toward tumor cells assessed by reactivity to the MuLV p15E-H2Kb tetramer (Fig. 4c, d). Given that MuLV p15E is an antigen expressed in MC38, Pan02, and KPC tumor cells while absent in host animals[36], we labeled Tc with MuLV p15E-H2Kb tetramer to determine the cell population that was tumor-specific.

Approximately, 30–50% of Tc in irradiated Sirpα$^{-/-}$ tumors 3 days post-IR were p15E-reactive, and among them, a significant fraction was CD44$^{+}$CD62L$^{-}$, indicating differentiation of Tc into effector memory T cells (T$_{EM}$). The tumor-specific (p15E-reactive) Tc and T$_{EM}$ persisted in irradiated Sirpα$^{-/-}$ mice and were readily detectable in mouse peripheral blood and the spleen two weeks after tumor eradication (Fig. 4e). Supporting this, irradiated MC38 tumors in Sirpα$^{-/-}$ mice also exhibited an increased frequency of Tc that recognize the MC38-specific tumor neo-antigen ADPGK[37] (Supplementary Fig. 6).

Notably, tumors infused with Sirpα$^{-/-}$ BMDMs in WT mice after IR also displayed similar robust expansion of GranzB$^{high}$p15E-reactive Tc, whereas those without infused Sirpα$^{-/-}$ BMDMs had poor induction of Tc (from 6 to 13%) that were mostly GranzB$^{low}$ and lacked tumor-specificity (non-p15E reactive) (Fig. 4f). Ex vivo Tc cytotoxicity assays further confirmed that Tc isolated from irradiated WT tumors were inert, while those from irradiated Sirpα$^{-/-}$ BMDM-infused tumors were highly cytotoxic and rapidly eliminated respective tumor cells at a low effector: target ratios (Fig. 4g, Supplementary Movie 1). The large expansion of tumor-specific Tc was critical for durable tumor regression, as depletion of Tc (by αCD8), but not CD4 T helper cells (Th; by αCD4), diminished the efficacy of RT in tumor-bearing Sirpα$^{-/-}$ mice (Fig. 4h).

**Activated Sirpα$^{-/-}$ macrophages preclude compensatory immunosuppression post-IR.** Further analyses of TME immune compartments revealed other prominent features synergistically augmenting Tc activity in irradiated Sirpα$^{-/-}$ mice while absent in irradiated WT mice. As summarized in Fig. 5a–c, notable changes in the post-IR Sirpα$^{-/-}$ TME included diminishment of CD4 Foxp3$^{+}$ Tregs but a significant increase in IFNγ$^{+}$ Th1 cells and NK cells. Despite maintaining similar total populations of intratumoral CD4 T cells (Th), irradiated tumors in Sirpα$^{-/-}$ mice exhibited a marked reduction of Foxp3$^{+}$ Tregs, which went from comprising >50% of the total Th cells to <10%; meanwhile, the IFNγ$^{+}$ Th1 population significantly expanded (Fig. 5a, b). This Th phenotypic switch (Treg → Th1) in the post-IR Sirpα$^{-/-}$ TME, along with other proinflammatory elements, collectively supports an immunogenic shift that favors tumor elimination. In contrast, irradiated tumors in WT mice failed to initiate the Th phenotypic switch and instead retained Tregs as the primary population. Furthermore, irradiated tumors in Sirpα$^{-/-}$ mice exhibited a fourfold increase in NK cells, which also expressed a high level of GranzB (Fig. 5c). This critical antitumor feature, however, was missing in WT mice.

It has been widely reported that RT-induced tumor damage can drive a strong wound-healing response characterized by the recruitment of monocytes, which function as monocytic MDSCs to suppress Tc while promoting tumor recovery and growth[38]. According to our gating strategy for analysis of the leukocyte population in the TME (Supplementary Fig. 7), we observed a similar phenomenon with irradiated MC38 tumors in WT mice having been highly infiltrated by Ly6C$^{high}$ monocytes/MDSCs (Fig. 5d). This compensatory pro-tumor mechanism post-RT, however, was explicitly absent in the Sirpα$^{-/-}$ TME. Intratumoral myeloid cells isolated from Sirpα$^{-/-}$ mice were significantly less inhibitory toward Tc proliferation than those from WT mice (Fig. 5d). In contrast, the post-RT Sirpα$^{-/-}$ TME contained significantly more proinflammatory Ly6G$^{high}$ PMNs, and the PMN obtained from the post-IR Sirpα$^{-/-}$ TME also exhibited characteristic proinflammatory features with high-level reactive oxygen species (ROS) (Fig. 5e). Significant infiltration of Tc and proinflammatory PMNs into irradiated tumors was observed in Sirpα$^{-/-}$ mice but not WT mice (Fig. 5f). We also found that

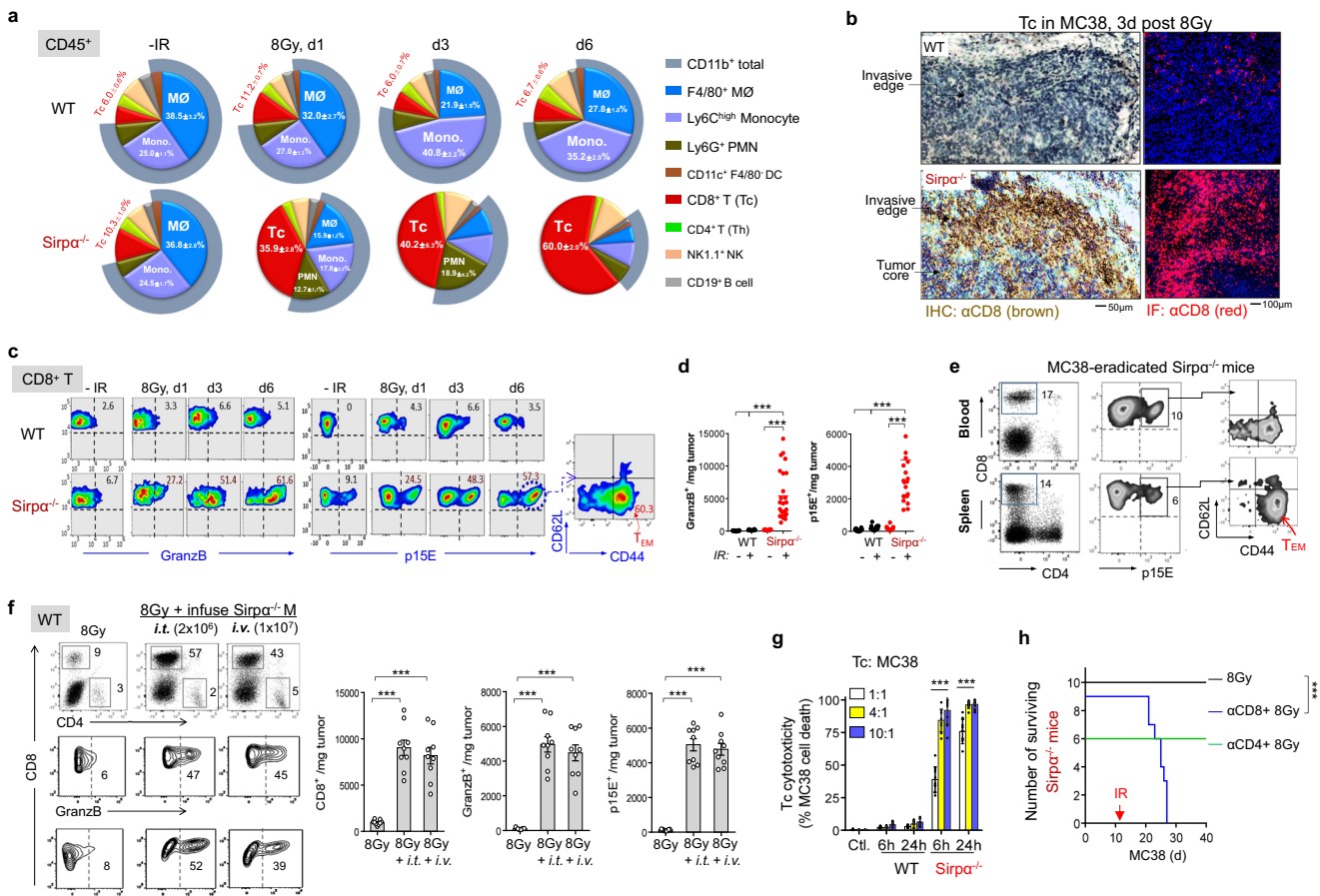

**Fig. 4 Sirpα deficiency induces robust tumor-specific Tc expansion following RT. a** Frequencies of intratumoral immune cell populations before and 3 days after an 8 Gy IR. Data from five independent experiments (3–4 mice/group in each experiment) are presented as mean ± SEM. $n = 16$ mice in each group. **b** IHC and IF staining of CD8+ Tc in MC38 tumors 3 days after IR. Data represent three independent experiments (3 mice/group in each experiment). **c, d**, Frequencies (**c**) and number (**d**) of granzyme B$^{high}$ (GranzB) and p15E+ Tc in the MC38 TME before and after IR. Data are representative of five independent experiments (3–5 mice/group in each experiment). $n = 25$ mice (Sirpα$^{-/-}$ + IR) and 16 mice (other groups). ***$P < 0.001$. **e** Frequency of p15E+ Tc and p15E+CD44+CD62L$^-$ T$_{EM}$ in peripheral blood and the spleen of MC38-eradicated Sirpα$^{-/-}$ mice. Data are representative of two independent experiments. $n = 6$ mice in each group. **f** MC38-bearing WT mice received two infusions (3-days interval) of Sirpα$^{-/-}$ BMDM via i.t. (1×10$^4$/ mm$^3$ tumor, total 2 × 10$^6$ for a 200 mm$^3$ tumor) or i.v. route (1 × 10$^7$ per mouse) followed by IR (8 Gy) after each infusion. Intratumoral p15E+ and GranzB+ Tc were quantified after 3 days. Data from three independent experiments (3 mice/group in each experiment) are presented as mean ± SEM. ***$P < 0.001$. $n = 9$ mice in each group. **g** Assaying Tc cytotoxicity in vitro. Tc from irradiated MC38 tumors of WT and Sirpα$^{-/-}$ mice were isolated 3 days post-IR and cocultured with MC38 cells at an indicated effector: target ratios for 6 or 24 h. Real-time Tc killing tumor cells are depicted in Supplementary Movie 1. Data from three independent experiments were presented as mean ± SEM. $n = 9$ mice in each group. ***$P < 0.001$. **h** Depletion of CD8 Tc (by αCD8) but not CD4 Th (by αCD4) diminished RT efficacy against MC38 in Sirpα$^{-/-}$ mice. Data represent three independent experiments. $n = 10$ mice (8 Gy), 9 mice (αCD8 + 8 Gy), and 6 mice (αCD4 + 8 Gy). ***$P < 0.001$. $P$ values were calculated by one-way ANOVA with Tukey's post hoc test (**d**, **f** and **g**) or one-way ANOVA with log-rank (Mantel–Cox) test (**h**). Source data are provided as a Source Data file.

extensive PMN infiltration in irradiated tumors in Sirpα$^{-/-}$ mice was positively correlated with rapid tumor elimination (Fig. 5g), supporting the tumoricidal role of these proinflammatory PMNs. Chemokine analysis (Supplementary Fig. 8a) and ex vivo chemotaxis assays (Supplementary Fig. 8b) corroborated the infiltration of different leukocytes in Sirpα$^{-/-}$ and WT TME, showing that irradiated tumors in Sirpα$^{-/-}$ mice produced and secreted a high level of CXCL1 (KC) that attracted PMN, whereas irradiated tumors in WT mice produced and secreted CCL2 (MCP-1) that mainly attracted monocytes or MDSC.

**Phagocytic Sirpα$^{-/-}$ macrophages present antigens to activate tumor-specific Tc in situ.** Given the predominant role of tumor-specific Tc in RT-induced tumor eradication in Sirpα$^{-/-}$ mice or in WT mice with Sirpα$^{-/-}$ macrophage infusion, we further

investigated how RT-activated intratumoral Sirpα$^{-/-}$ macrophages drive the robust expansion of tumor-specific Tc. The high expression of immunogenic antigen presentation machinery on intratumoral Sirpα$^{-/-}$ macrophages after IR suggests that, following phagocytosis of tumor cells/debris, Sirpα$^{-/-}$ macrophages may function as antigen-presenting cells to activate tumor-specific Tc. Given the large scale and rapid expansion of intratumoral Tc in irradiated Sirpα$^{-/-}$ mice, we postulated that the tumor antigen presentation by Sirpα$^{-/-}$ macrophages occurred in situ and led to a local anamnestic response via 'calling' tumor-specific memory T cells (i.e., T$_{EM}$ and T$_{RM}$)[39,40]. Two lines of experiments were performed to test this hypothesis. First, immediately after IR (15 min post 8 Gy), tumors were excised without tumor-draining lymph nodes (TDLN) and cultured ex vivo (Fig. 6a). Despite the absence of the TDLN, irradiated tumor explants from Sirpα$^{-/-}$ mice exhibited robust expansion of Tc similar to that which

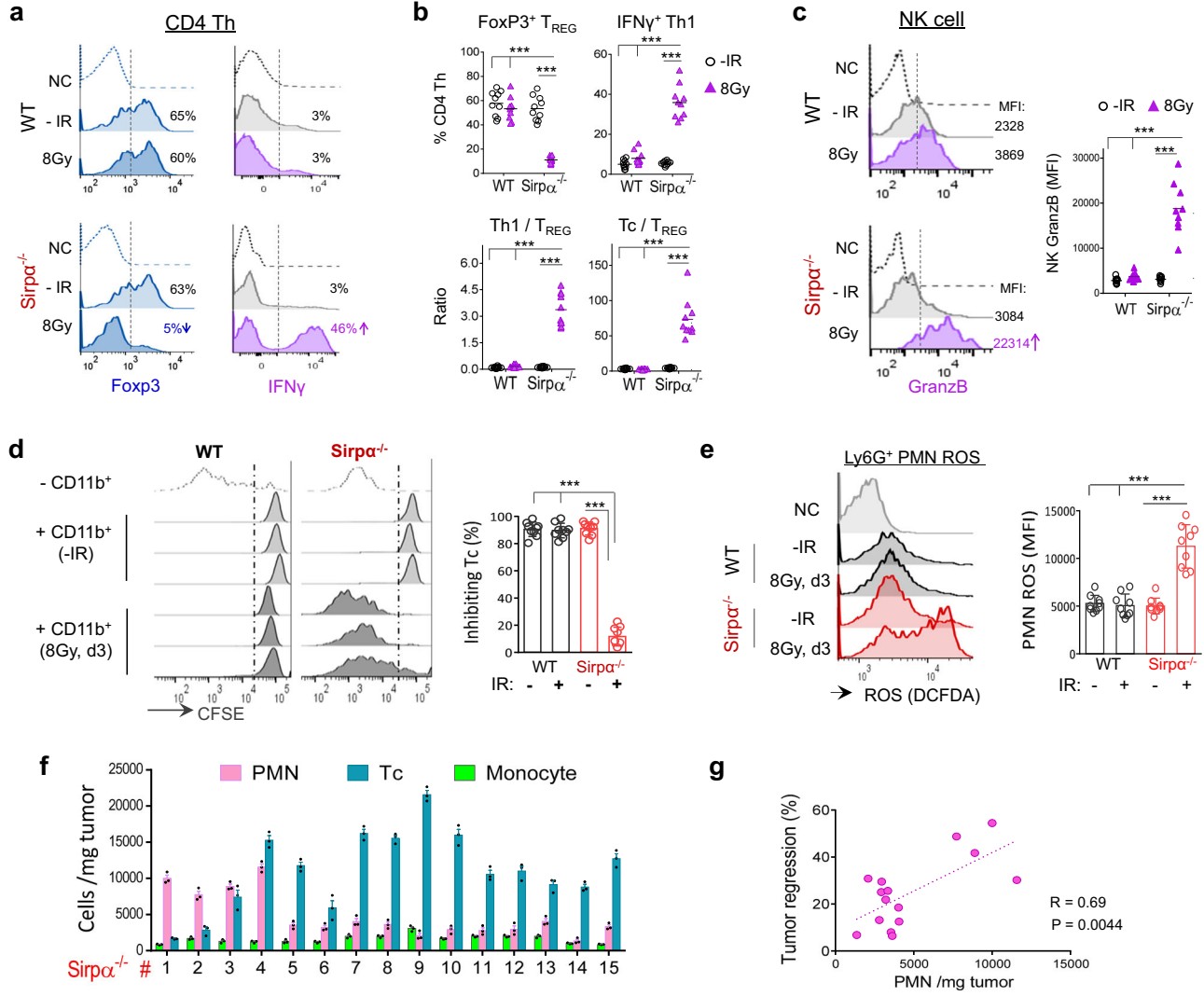

**Fig. 5 Reshaping the immunosuppressive TME in Sirpα$^{-/-}$ mice into a proinflammatory tumoricidal niche post-IR. a, b** Foxp3$^+$ Tregs and IFNγ$^+$ Th1 among the total intratumoral CD4 Th cells. Data are representative of three independent experiments (3–4 mice/group in each experiment). $n = 10$ mice in each group. ***$P < 0.001$. **c** NK cells and their GranzB expression. Left, representative image from three independent experiments showing the mean fluorescent intensity (MFI) of GranzB in intratumoral NK cells. Right, statistical analysis of GranzB MFI in intratumoral NK cells). ***$P < 0.001$. **d** Isolated total intratumoral myeloid cells were tested for inhibition of T cell proliferation. **e** ROS production by tumor infiltrated PMN in the presence of DCFDA and 1 μM PMA. Data from three independent experiments are presented as mean ± SEM. $n = 9$ mice in each group (**c–e**). ***$P < 0.001$. **f, g** Infiltrated PMNs promote tumor regression in Sirpα$^{-/-}$ mice. Large quantities of intratumoral PMN and other leukocytes were detected in most of the fifteen MC38 tumors in Sirpα$^{-/-}$ mice collected 3 days post-IR (**f**), and intratumoral PMN quantity positively correlated with the degree to which tumors had regressed by 3 days post-IR (**g**). Cell numbers were counted in triplicate and presented as mean ± SEM. Data are representative of three independent experiments. $n = 15$ mice. ***$P < 0.001$; one-way ANOVA with Tukey's post hoc test (**b**, **c**, **d**, and **e**). For Fig. 5g, the dot line corresponds to the best linear fit to the number of PMN and tumor regression. Pearson correlation coefficient ($R$) and $P$ value were calculated using linear regression with 95% confidence interval. Source data are provided as a Source Data file.

occurred in vivo (Fig. 4a). Infusing Sirpα$^{-/-}$ BMDM into irradiated tumors from WT mice after their excision also induced significant Tc expansion, whereas those tumors without Sirpα$^{-/-}$ macrophage infusion exhibited no Tc expansion. These results suggest that Sirpα$^{-/-}$ macrophages activated by irradiated tumor-released DAMPs drive Tc expansion in situ independent of peripheral lymphoid organs.

Second, an in vitro coculture system of Sirpα$^{-/-}$ macrophages with tumor-infiltrating lymphocytes (TILs) was established. As depicted in Fig. 6b, Sirpα$^{-/-}$ BMDMs were first incubated with irradiated MC38 or PDA tumor dissociates for activation and phagocytosis of tumor antigens. After 18 h, the tumor antigen-loaded Sirpα$^{-/-}$ BMDMs, which by then also exhibited proinflammatory characteristics and increased immunogenic

antigen presentation machinery (Supplementary Fig. 9a–c), were cocultured with TILs isolated from the mice bearing the same type of tumor. As shown in Fig. 6c, tight engagements between activated Sirpα$^{-/-}$ BMDMs and tumor-specific Tc were seen within 1 h of co-incubation, in a fashion reminiscent of mature dendritic cells forming antigen-presenting conjugates with T cells[41]. We noticed that the Tc engaged with Sirpα$^{-/-}$ BMDMs represented less than 5% of the total applied TILs, which is in agreement with the previous finding that the majority of TILs were irrelevant or "bystander" T cells[42]. Interestingly, activated Sirpα$^{-/-}$ BMDMs did not phagocytose their engaged Tc to which they putatively presented tumor antigens. Following 1–3 days of coculture, these Sirpα$^{-/-}$ BMDM-engaged Tc still displayed cellular activation (i.e., blasting) (Fig. 6d) and robust proliferation

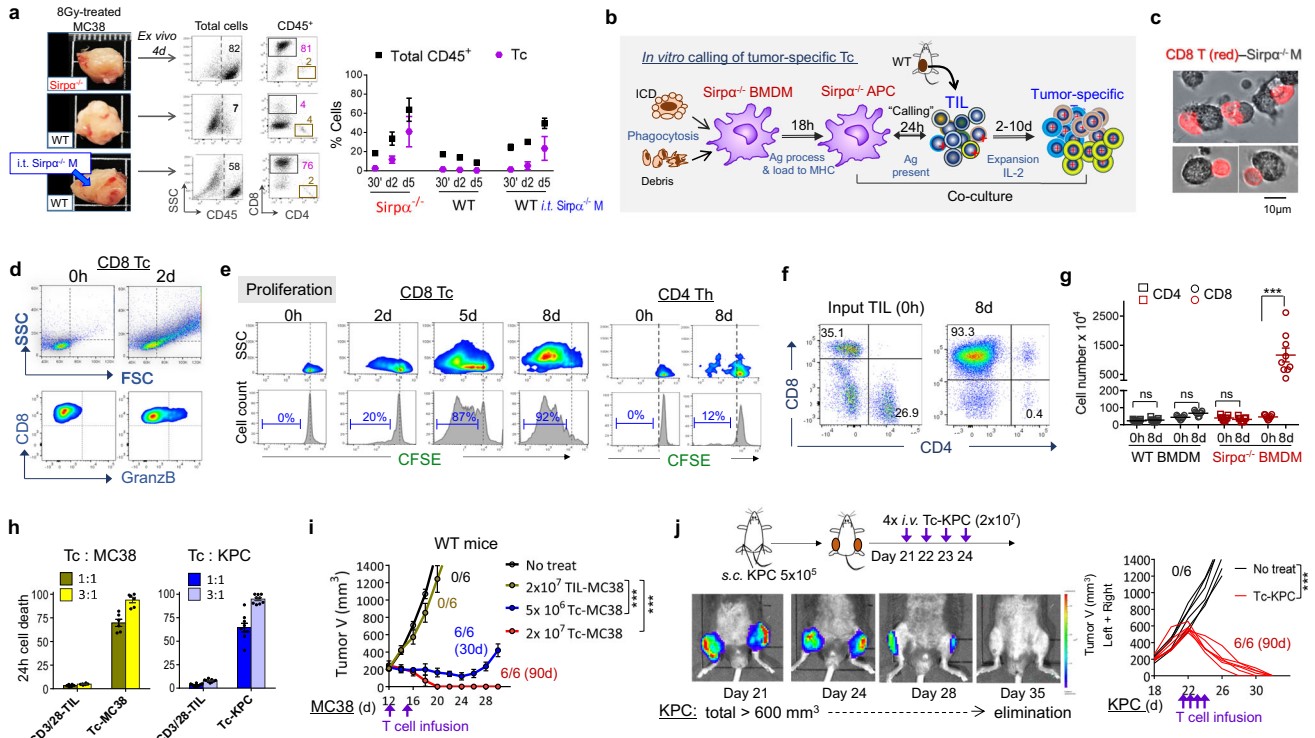

**Fig. 6 Phagocytic Sirpα−/− macrophages function as APC and activate tumor-specific Tc. a** After IR (8 Gy), MC38 tumors (~300 mm³) were excised, digested, and cultured ex vivo; some tumors from WT mice were i.t. injected with 3 × 10⁶ Sirpα−/− BMDMs prior to cutting and culture. Single-cell suspensions were prepared at indicated time-points and analyzed for Tc and Th in CD45⁺ population. Data are representative of three independent experiments. n = 9 mice in each group. Each tumor was tested in triplicate. Data are presented as mean ± 95% confidence interval. **b** Experimental scheme of in vitro expansion of tumor-specific Tc from TILs by tumor-phagocytosed Sirpα−/− BMDMs. **c** Images of Tc (red, CD8 staining) from MC38 tumor-forming conjugates with MC38 antigen-loaded Sirpα−/− BMDMs (gray). Data represent three independent experiments. **d–g** Specific activation of Tc by tumor antigen-loaded Sirpα−/− BMDMs. Activation of Tc after 2–8 days of coculture of TILs with tumor antigen-loaded Sirpα−/− BMDMs was evident by cell size enlargement (**d**) and robust proliferation (**e–g**) of Tc but not Th. FACS data (**d–f**) are representative of three independent experiments coculturing MC38 TIL with MC38 tumor antigen-loaded Sirpα−/− BMDMs. Number of T cells before and after coculture (**g**) was presented as mean ± SEM and represent three independent experiments. WT BMDMs were used as control. n = 9 in each group. ***P < 0.001. For each experiment, TILs from 3 to 4 mice were pooled for expansion. Similar results were also observed in experiments coculturing KPC TIL with KPC tumor antigen-loaded Sirpα−/− BMDMs. **h** Tumoricidal activity of Tc expanded by M38- or KPC-loaded Sirpα−/− BMDMs compared to respective TILs nonspecifically expanded by αCD3/CD28. Expanded Tc or TILs were added into MC38 or KPC cell culture at indicated effector:target ratios for 24 h. Experiments were performed in triplicate and data from three independent experiments are presented as mean ± SEM. For the testing killing of MC38 cells, n = 6 per group. For the testing killing of KPC cells, n = 9 per group. **i, j** Adoptive T-cell therapy testing Tc-MC38 and Tc-KPC tumoricidal activity in vivo. WT mice bearing MC38 (**i**) or KPC (**j**) tumors received Tc-MC38 or Tc-KPC, or the same number of αCD3/CD28-expanded TILs at indicated time points. Tumor volume and animal survival were monitored for 90 days post-treatment. Data from two independent experiments (3 mice/group in each experiment) are presented as mean ± SEM (**i**) or individual tumor volume (**j**). ***P < 0.001. P values were calculated by one-way ANOVA with Tukey's post hoc test (**g, h, i**, and **j**). Source data are provided as a Source Data file.

(Fig. 6e). Despite that the initial TILs comprised both Tc and Th cells, Sirpα−/− BMDMs exclusively induced robust proliferation of Tc but not Th cells, a phenomenon mirroring tumor-specific Tc expansion in irradiated Sirpα−/− tumors in vivo. As shown in Fig. 6f–g, after 8 days of expansion in the presence of IL-2, large numbers of Tc were obtained.

In vitro cytotoxicity assays ascertained the high cytotoxicity of Sirpα−/− BMDM-expanded Tc, which induced rapid cancer cell death (Fig. 6h, Supplementary Movies 2 and 3) at low effector: target ratios (1–3:1). Interestingly, these expanded Tc comprised only a small fraction (<5%) of clones that were reactive toward the tumor-specific antigen p15E (Supplementary Fig. 9d), suggesting that the tumor-specific Tc population comprised other tumor-reactive clones that recognized other tumor-specific antigens. We next assessed the functional capacity of these Tc to eliminate tumors in vivo. In these experiments, Sirpα−/− BMDM-expanded Tc specific for MC38 or KPC (termed Tc-MC38 or Tc-KPC) were i.v. administered to WT mice bearing the same

tumor, respectively. As shown in Fig. 6i–j, Tc-MC38 or Tc-KPC dose-dependently killed MC38 or KPC tumors in WT mice. Two rounds of Tc-MC38 infusion, with each dose containing 2 × 10⁷ Tc, completely eliminated s.c engrafted MC38 tumors larger than 200 mm³; four rounds of Tc-KPC infusion cleared multiple KPC tumors whose collective volume exceeded 600 mm³. For comparison, parallel infusions of TILs expanded by antibody-ligation of CD3 and CD28, a nontumor-specific method[38], were ineffective against the same-sized tumors. Collectively, these results confirmed that Sirpα−/− macrophages alone are sufficient to induce tumor-specific Tc expansion in situ in irradiated tumors.

**Irradiated Sirpα−/− mice exhibit abscopal tumor remission and long-lasting immunity**. In rare instances, RT drives an immune response robust enough to control tumor burden outside the irradiated area, i.e., abscopal effect[5]. Given that RT-activated Sirpα−/− macrophages robustly induce tumor-specific cytotoxic

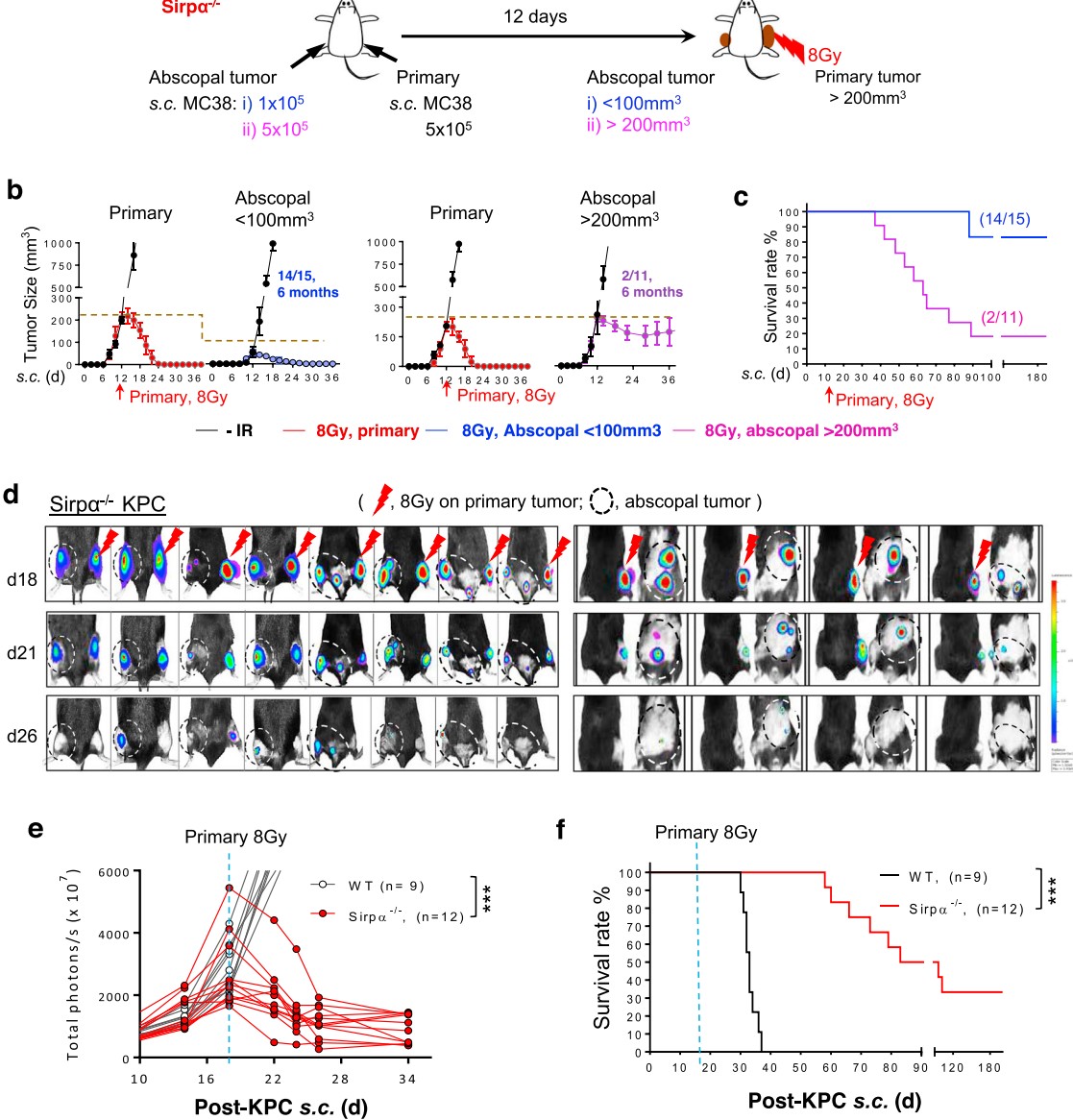

**Fig. 7 Sirpα$^{-/-}$ mice exhibit RT-induced abscopal effects and long-lasting anti-tumor immunity. a–c** MC38 tumors were simultaneously engrafted in the right flank (5 × 10$^5$, primary) and left flank (1–5 × 10$^5$, abscopal) in Sirpα$^{-/-}$ mice. The primary tumor on the right flank was irradiated with 1 × 8 Gy on day 12 post tumor engraftment when the volume of abscopal tumors was <100 mm$^3$ or >200 mm$^3$ (**a**). Tumor volume (**b**) and animal survival (**c**) were recorded. Data from three independent experiments (3–5 mice/group in each experiment) are presented as mean ± SEM. $n = 15$ mice (abscopal < 100 mm$^3$) and 11 mice (abscopal > 200 mm$^3$). **d–f** Luciferase-expressing KPC tumors were simultaneously engrafted in the right flank (primary) and other locations including left flank, dorsal or peritoneal cavities in WT and Sirpα$^{-/-}$ mice, followed by irradiation on the primary tumor once its volume exceeded 200 mm$^3$. Representative images show the responses and abscopal effects of luciferase-expressing KPC tumors to IR in Sirpα$^{-/-}$ mice. Analysis of tumor development using IVIS imaging system (**e**) and animal survival (**f**) was recorded. Data are representative of three independent experiments (3–4 mice/group in each experiment). $n = 9$ mice (WT) and 12 mice (Sirpα$^{-/-}$). ***$P < 0.001$. $P$ values were calculated by one-way ANOVA with Tukey's post hoc test (**e**) or one-way ANOVA log-rank (Mantel–Cox) test (**f**). Source data are provided as a Source Data file.

CD8 T cells, we further tested whether IR of primary lesions in Sirpα$^{-/-}$ mice could have an abscopal effect on nonirradiated tumors. In this experiment, MC38 or KPC tumors were engrafted in both the right and left flank. When the tumor in the right flank (the primary tumor) reached >200 mm$^3$, a single 8 Gy IR was given locally (Fig. 7a). KPC tumors were also engrafted in the dorsal area and/or the peritoneal cavity. As shown in Fig. 7b–f and Supplementary Fig. 10, IR of the primary tumor in Sirpα$^{-/-}$ mice not only led to its elimination but also induced rapid regression of nonirradiated tumors in the left flank (Fig. 7b, c), the dorsal and/or the peritoneum (Fig. 7d–f), though the

regression of abscopal tumors was slower compared to that of irradiated primary tumors. We also observed that when non-irradiated tumors were large (>200 mm$^3$), a single dose of 8 Gy IR administered to the primary tumor failed to control the growth of non-irradiated tumors in Sirpα$^{-/-}$ mice. To explore the difference in these responses, we analyzed the TME in large (>200 mm$^3$) and small (<100 mm$^3$) nonirradiated tumors. The results showed significantly greater infiltration of CD8 Tc into small nonirradiated tumors than large nonirradiated tumors, supporting the role of Tc in the abscopal effect on small tumors (Supplementary Fig. 11a). In contrast, the TME of large

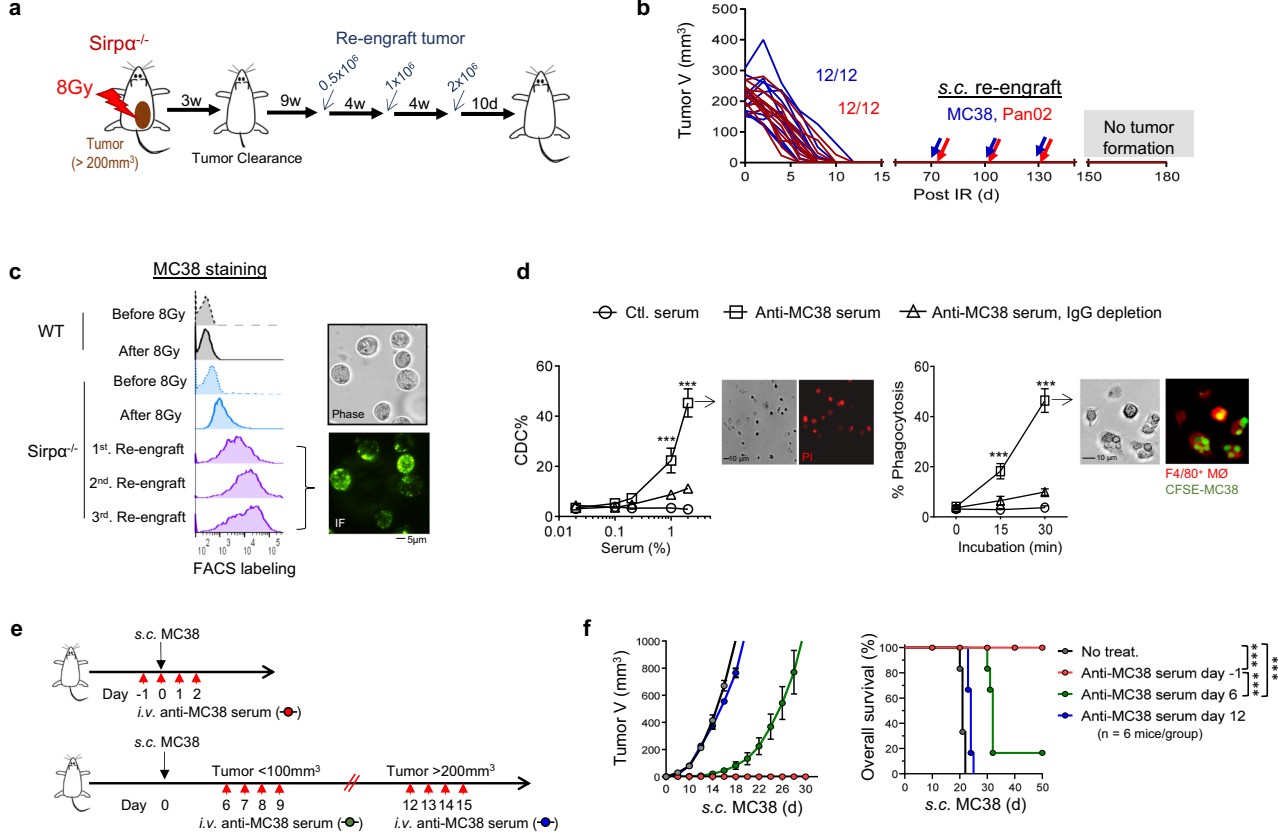

**Fig. 8 Long-lasting humoral antitumor immunity in tumor-eradicated Sirpα$^{-/-}$ mice. a** Experimental scheme. After eradicating MC38 or Pan02 tumors (3-weeks post-RT), Sirpα$^{-/-}$ mice were re-challenged with three rounds of dose-escalating inoculums of the same cancer cells. **b** Tumor size post-RT and MC38 or Pan02 re-engraftment. Data are representative of three independent experiments (4 mice/group in each experiment). $n = 12$ mice in each group. **c** Immunostaining of MC38 tumor cells using serum from MC38-eradicated Sirpα$^{-/-}$ mice in combination with fluorescent-conjugated antibody against murine IgG. Data are representative of three independent experiments (4 mice/group in each experiment). $n = 12$ mice in each group. **d** Anti-MC38 serum-induced CDC (left panel) and Fc-mediated phagocytosis (right panel) of MC38 cells. Data are representative of two independent experiments with 3 mice/group in each experiment are presented as mean ± SEM. $n = 6$ mice in each group. Experiments were performed in duplicate. ***$P < 0.001$. **e**, **f** Preventative but not therapeutic effect of anti-MC38 serum. WT mice received anti-MC38 serum treatment (200 μl/mouse, *i.v.*) prior to or after MC38 engraftment at various time points as indicated (**e**). Tumor growth and animal survival (**f**) were recorded. Data from two independent experiments (3 mice/group in each experiment) are presented as mean ± SEM. $n = 6$ mice in each group. ***$P < 0.001$. $P$ values were calculated by one-way ANOVA with Tukey's post hoc test (**d**) or one-way ANOVA log-rank (Mantel–Cox) test (**f**). Source data are provided as a Source Data file.

nonirradiated tumors was still dominated by pro-tumor myeloid cells. We further analyzed the TAMs that remained in the large nonirradiated tumors that were not eliminated by the abscopal effect. As shown in Supplementary Fig. 11b, unlike TAMs in irradiated tumors, which displayed significantly increased antigen presentation capacity and M1-phenotypic activation, TAMs in nonirradiated tumors retained a lower expression of antigen presentation machinery MHC-I, MHC-II, CD80, CD86, OX40L, M1-phenotypic marker iNOS but a higher level of M2-phenotypic marker CD206.

**Tumor-eradicated Sirpα$^{-/-}$ mice acquire humoral antitumor immunity.** In MC38 or Pan02 tumor-eradicated Sirpα$^{-/-}$ mice, we performed tumor re-engraftment to determine whether long-term antitumor immunity had manifested (Fig. 8a). As shown, a dose-escalating amount of MC38 or Pan02 cells was re-engrafted into MC38 or Pan02 tumor-eradicated Sirpα$^{-/-}$ mice. Injected cancer cells failed to grow in tumor-eradicated Sirpα$^{-/-}$ mice (Fig. 8b), suggesting that these mice had acquired long-term antitumor immunity that prevented tumor recurrence. Moreover, the sera from this MC38 tumor-eradicated Sirpα$^{-/-}$ mice directly labeled MC38 cells (Fig. 8c), suggesting potential acquisition of humoral antitumor immunity in these tumor-eradicated mice.

Furthermore, in vitro assays showed that these sera mediated cancer cell killing through complement-dependent cytotoxicity (CDC) (Fig. 8d, left) and Fc-mediated phagocytosis (Fig. 8d, right).

The antitumor efficacy of the sera from tumor-eradicated mice was further tested in WT recipient mice. As shown in Fig. 8e, sera were administered multiple times in three ways: prior to and immediately following tumor engraftment; at the time point when engrafted tumors had grown to less than 100 or over 200 mm³. Injection of the sera from tumor-eradicated Sirpα$^{-/-}$ mice successfully prevented nascent tumor formation in WT recipient mice (Fig. 8f). However, the sera from tumor-eradicated Sirpα$^{-/-}$ mice only delayed the growth of small tumors (<100 mm³) and failed to control the growth of large tumors (>200 mm³) in WT recipient mice (Fig. 8f). These results suggest that the sera from tumor-eradicated Sirpα$^{-/-}$ mice have the capacity to prevent nascent tumor formation but do not elicit a robust therapeutic effect that controls pre-established tumors.

## Discussion

As the most common immune cell in the TME, intratumoral macrophages play an important role in cancer development and are often correlated with poor prognosis and therapy resistance,

including immunotherapies. An elegant study by John et al.[21] demonstrated that simply depleting macrophages could enhance the efficacy of antitumor RT. However, here we demonstrate that the presence of Sirpα-deficient intratumoral macrophages can instead augment cancer therapies, including RT. Our study unveils SIRPα as a master controller of immunity in the TME by directing the post-treatment response toward wound-healing, strengthening immunosuppression, conferring treatment resistance, and ensuring tumor progression. Conversely, antitumor immune responses are significantly enhanced in the absence of SIRPα. Unlike normal tumor-associated macrophages, which exert high pro-tumor activity, activated Sirpα-deficient intratumoral macrophages exhibit a superior antitumor capacity and function as proinflammatory phagocytes and immunogenic APC that transform the TME into a tumoricidal niche highly infiltrated by tumor-killing cytotoxic T cells, NK cells, and inflammatory neutrophils, but with limited immunosuppressive cytokines, Tregs and MDSC. The presence of intratumoral SIRPα-deficient macrophages alone appears to subvert the paradigmatic immunosuppressive response to RT and instead empower IR to eliminate otherwise treatment-resistant, late-stage colorectal, and pancreatic cancers, achieving up to 100% survival rates and long-lasting humoral and cellular immunity that effectively prevent a recurrence. Supporting the role of Sirpα-deficient macrophages in activating cancer-specific Tc to eliminate tumors, Sirpα-deficient macrophages plus local IR also induce abscopal remission of smaller tumors (<100 mm³). Indeed, the augmented anti-cancer efficacy of RT conferred by Sirpα-deficient macrophages was unmatched by therapeutic combinations with anti-PD-L1 checkpoint inhibitors, CD47-blockade, and/or anti-tumor antibodies.

How does a single protein, its expression or deficiency, so profoundly impact the manner in which tumors respond to RT? Indeed, SIRPα mediates inhibition that not only restrains macrophages from phagocytosis of tumor cells but also dampens their capacity to exert proinflammatory responses and immunogenic antigen presentation that drive antitumor adaptive immunity. SIRPα in intratumoral macrophages thus steers the RT-induced immune response within the TME toward immunosuppression and wound healing. As shown in this study, responding to DAMPs in the context of RT, intratumoral macrophages expressing SIRPα rapidly increased the production and secretion of TGFβ and IL-10 and chemokines to recruit MDSC, together with strengthening an immunosuppressive TME and initiating wound-healing to promote tumor progression. Substantial increases of three TGFβ isoforms (TGFβ1–3) following IR of MC38 or PDA tumors in WT mice are in line with previous reports by others, who showed that TGFβ blockade improves the efficacy of RT in preclinical models and clinical settings[35,43]. At this stage, it remains unknown how SIRPα mediates such a strong immunosuppressive response in the TME to bolster tumor resistance to RT. Our finding that CD47 blockade failed to recapitulate the phenomenon of tumor eradication by RT-activated Sirpα−/− macrophages, however, strongly argue that SIRPα controls tumor responses to RT via a CD47-independent pathway. Therefore, further investigation is required to unravel the multifaceted capacity of SIRPα that regulates macrophages and thereby drives the entire tumor-immune response to resist RT.

We show that the presence of activated Sirpα−/− macrophages in tumors alone shifts the post-radiation response of TME from pro-tumor wound-healing to tumor elimination. In fact, the mere depletion of SIRPα without RT is ineffective, as evidenced by Sirpα−/− mice exhibiting neither inherent autoimmunity nor innate resistance to tumor engraftment and growth. Likewise, WT mice infused with Sirpα−/− macrophages in tumors did not curtail

tumor progression in the absence of RT. However, SIRPα deficiency combined with macrophage activation, herein achieved by DAMPs released by irradiated tumor cells, transformed intratumoral Sirpα−/− macrophages into exceptional antitumor effector cells that not only phagocytose tumor cells but also propagate the anti-tumor response by presenting tumor antigens to activate tumor-specific Tc. In addition to activating adaptive immune components against cancer, activated intratumoral Sirpα−/− macrophages by irradiated tumor cell-released DAMPs also elicit a robust proinflammatory response that reshapes the TME to favor tumor elimination. As shown by Nanostring and flow cytometric analyses, activated intratumoral Sirpα−/− macrophages following local radiation exhibited a high production of proinflammatory cytokines and chemokines that recruit tumoricidal PMNs and T cells, and also elevated the expression of immunogenic antigen presentation machinery including MHC-I and MHC-II, as well as an array of co-stimulatory molecules such as CD80, CD86, and OX40L. Consequently, the drastic changes in the TME brought forth by activated Sirpα−/− macrophages culminated in rapid tumor elimination via large numbers of tumoricidal Tc and NK cells, infiltration of proinflammatory PMN, and a simultaneous reduction of immunosuppressive components including Treg, MDSC, and inhibitory cytokines such as TGFβ. Considering the fact that RT-activated Sirpα−/− macrophages possess a highly immunogenic antigen presentation capacity, we have established an in vitro system to demonstrate that Sirpα−/− macrophages may present tumor antigens to specifically activate tumoricidal Tc. These experiments, which build upon the elegant framework previously established by others exploring ex vivo expansion of TILs[44], confirm that phagocytic Sirpα−/− macrophages are apt to immunogenically 'call' tumor-specific memory T cells in situ, thus explaining the kinetics and breadth of Tc expansion observed in irradiated tumors in Sirpα−/− mice. The success of this in vitro system also provides a foundation enabling the proficient expansion of highly cytotoxic tumor-specific T cells that may be deployed in potent tumor-eliminating adoptive cell therapies. Immune checkpoint inhibition, especially PD-L1 blockade, has been successfully applied to amplify T cell anti-tumor immunity, with this strategy having been employed alone[45] or in combination with RT[21,46]. Given that robust proliferation and activation of antitumor Tc is integral to tumor eradication induced by Sirpα−/− macrophages and local RT, implementing PD-L1 blockade into this combination may significantly enhance the efficacy of Sirpα−/− macrophage-based antitumor treatments.

In summary, the present study unveils an important mechanism by which malignant tumors resist radiotherapy and other tumor-killing immunomodulatory regimens. This mechanism is mediated by intratumoral macrophages via its inhibitory regulator SIRPα and depleting intratumoral macrophage SIRPα empowers radiotherapy and other immunotherapies to eliminate large colorectal and pancreatic tumors, induce abscopal remission and confer long-lasting antitumor immunity. Our study provides proof-of-principle for developing Sirpα−/− macrophage-based therapies to cure a broad spectrum of cancers including those at advanced stages with metastasis.

## Methods

**Cell lines**. MC38 and Pan02 cells were obtained from NCI (Developmental Therapeutics Program). KPC-luc cells were kindly provided by Dr. Edmund Waller (Emory University). All cell lines were cultured in DMEM supplemented with 10% fetal bovine serum (FBS).

**Mice and syngeneic tumor models**. All experiments using animals and procedures of animal care and handling were carried out following protocols approved by the Institutional Animal Care and Use Committee (IACUC) of Georgia State University. Age- and sex-matched male and female adult (8–16-week-old) Sirpα−/− or Sirpα−/−-GFP mice and their wild-type (WT) littermates were used in each independent experiment. The generation and

characterization of Sirpα$^{-/-}$ mice have been described previously[11]. Sirpα$^{-/-}$-GFP mice were generated by crossing Sirpα$^{-/-}$ mice with C57BL/6-Tg (UBC-GFP) 30Scha/J (The Jackson Laboratory). Mouse were housed in an institutional pathogen-free facility on a 12-h reverse light/dark cycle (7PM–7AM). The animal facility was maintained at a temperature of 23 ± 2 °C with 55 ± 5% humidity. To establish primary tumors, 5 × 10$^5$ MC38, Pan02 or KPC-luc syngeneic cancer cells in 50 μl phosphate-buffered saline (PBS) were subcutaneously (s.c.) engrafted into the right flank of mice. To determine whether the abscopal effect could occur, 1–5 × 10$^5$ cancer cells were simultaneously engrafted (s.c.) into the left flank, dorsal area, or peritoneal cavity. Tumors were measured every other day after their formation using calipers and tumor volume was calculated using the formula: volume = (length × width$^2$)/2. To measure KPC-luc tumor growth by bioluminescence imaging, mice were anesthetized (inhalational isoflurane), followed by intraperitoneal (i.p.) injection with D-luciferin (100 mg/kg, Promega) and imaging using an IVIS Spectrum in vivo imaging system and Living Image software (Perkin Elmer). The humane endpoints were determined by the tumor size (individual tumor reached 1500 mm$^3$) and/or animal discomfort level (displayed squinting, ungroomed, hunched, and/or it does not right itself when placed on its side). In separate experiments, tumor-eradicated mice were re-challenged with the same cancer cells at different time points to test tumor resistance.

**Radiotherapy**. All single-fraction RT treatments in this study were applied directly to established primary tumors when their volume reached the designated range ("small": <100 mm$^3$; "medium": 100–400 mm$^3$; "large": 400–600 mm$^3$) or exceeded 200 mm$^3$. To perform RT, tumor-bearing mice were first anesthetized with ketamine (17.5 mg/ml, Henry Schein) and xylazine (2.5 mg/ml, Henry Schein), and were then placed in a customized jig with a lead holder such that only the primary tumor was exposed, followed by irradiation in an RS-2000 biological X-ray irradiator (Rad source technology, Suwanee, GA) with a dose rate of 1.2 Gy/min (160 kV, 25 mA) to reach 4 Gy, 8 Gy or 15 Gy. Mice were monitored post-RT for discomfort and change in tumor volume. Mice that had eradicated their tumors following RT were maintained for an additional 6 months to 1.5 years to monitor tumor recurrence and determine long-term survival rates. For RT combination modalities, primary tumors in WT mice prior to (<3 h) or immediately after IR were intratumorally (i.t.) administered 50 μg rat-anti-mouse CD47 (αCD47, clone miap301), or soluble murine SIRPα extracellular domain fusion protein (mSirpα.ex, also termed mSirpα.ex-Fc). Rat IgG isotype, and soluble human SIRPα extracellular domain fusion protein (hSirpα.ex)[47]. To test RT combined with Sirpα$^{-/-}$ macrophage infusion, macrophages derived from Sirpα$^{-/-}$ mouse bone marrow (BMDM) were generated in vitro. Briefly, bone marrow cells were obtained by flushing both femur bones with 1-2 ml of ice-cold PBS. After lysis of RBC using RBC lysis buffer (0.42 g NH$_4$Cl, 0.05 g NaHCO$_3$, 0.05 g EDTA in 50 ml cell culture grade water), cells were then placed in culture with RPMI-1640 containing 10% FBS, 1% penicillin/streptomycin, and M-CSF (10 ng/ml, Peprotech) for 5–6 d to generate BMDMs. Prior to or after RT, Sirpα$^{-/-}$ BMDM were adoptively transferred via i.t. (0.2–1 × 10$^4$ cells/mm$^3$ tumor volume) or i.v. (0.5–2 × 10$^7$ cells/mouse) into tumor-bearing WT mice. In some cases, the recipient mice received the same Sirpα$^{-/-}$ BMDM transfusion three days later. Biofluorescence imaging (IVIS Spectrum, Perkin Elmer) was used to trace the biodistribution of i.v. administered GFP-Sirpα$^{-/-}$ BMDM in tumor-bearing mice. To determine critical immune populations underlying RT efficacy, depletion of intratumoral macrophages or T cells was performed in tumor-bearing Sirpα$^{-/-}$ mice. To deplete intratumoral macrophages, clodronate-containing liposomes (Cl2MDA-liposomes) or rat-anti-mouse CSF1R antibodies (clone AFS98) were i.t. injected (100 μg/tumor) in 50 μl PBS using a 30 G Ultra-Fine Insulin Syringe (BD). To deplete CD8 or CD4 T cells, rat-anti-mouse CD8α (clone 2.43) or CD4 (clone GK1.5) antibodies were used (100 μg/tumor, i.t.). These liposome or antibody treatments were conducted twice (2×, 2 days and 3 h) prior to IR. Liposomes without clodronate and Rat IgG isotype were used as controls.

**Serum cytokine detection**. Prior to (0 h) and after (24 and 72 h) tumor IR, the tumor-bearing WT, and Sirpα$^{-/-}$ mice were exsanguinated to obtain serum. Cytokines including TNFα, IL-1β, IL-12, and IFNγ in mouse sera were detected by a multiplex assay (BioLegend) according to the manufacturer's instructions.

**Antitumor serum analyses**. To test if tumor-resistant Sirpα$^{-/-}$ mice developed humoral immunity, serum samples were collected from these mice 10 days after the third round of tumor re-engraftment. The sera were used at dilutions of 1:200, 1:500, 1:1000, 1:2000, 1:5000, 1:10,000, and 1:20,000 for cell surface staining of the corresponding tumor cells cultured in a 24-well plate (30 min, 4 °C), followed by detection with an Alexa488-conjugated rat anti-mouse IgG secondary antibody. After washing, the labeled cells were examined by microscopy and read by flow cytometry. To assay CDC, serum samples, without and with immunoglobulin depletion by protein A/G-conjugated Sepharose, were added at dilutions of 1:50, 1:100, 1:500, 1:1000, and 1:5000 into cultured cancer cells (10$^5$ cells/well) followed by incubation at 37 °C in the presence of propidium iodide (PI). After 30 min, cancer cell death was examined by fluorescent microscopy and flow cytometry [% CDC = 100 × (PI$^+$ cancer cells/100 cancer cells)]. To assay antibody/Fc-mediated phagocytosis, serum samples were added (1:500 dilution) into WT

BMDM (10$^5$ cells/well) culture in a 24-well plate along with corresponding cancer cells (5 × 10$^5$ cells/well) pre-labeled with carboxyfluorescein succinimidyl ester (CFSE). After 15 or 30 min (37 °C), cells were gently washed and then were labeled with PE-conjugated anti-F4/80 antibody. Macrophage (red) phagocytosis of CFSE-labeled cancer cells (green) was assessed by fluorescent microscopy and flow cytometry [% phagocytosis = 100 × (macrophages that ingested ≥ 1 target/100 macrophages)]. To assess the efficacy of anti-tumor serum in vivo, WT mice were i.v. injected with antitumor serum (undiluted, 200 μl) once a day for four consecutive days either prior to (day −1) or after (day 6 or day 12) s.c. engraftment of MC38 (5 × 10$^5$ cells). Comparable experiments and controls were done using sera from healthy WT and Sirpα$^{-/-}$ mice, as well as from tumor-bearing mice prior to and at various times after receiving RT.

**TME analysis**. Tumors were excised without draining lymph nodes and were weighed. For immunohistochemistry (IHC) analyses, tumors were embedded in Richard-Allan Scientific Neg-50 frozen section medium (Thermo Scientific) followed by sectioning at a thickness of 10 μm using a cryostat microtome. To perform IHC staining, the slides were treated with 0.1% H$_2$O$_2$, blocked with 10% bovine serum albumin (BSA) in PBS, followed by staining for CD8 T cells and other leukocytes using biotin-conjugated rat anti-mouse CD8a (BioLegend) or other primary antibodies (4 °C, 18 h). After washing, the slides were further incubated with HRP-conjugated streptavidin followed by color development using 3,3′-diaminobenzidine (both from Thermo Fisher Scientific). Tissue sections were then counterstained with hematoxylin and mounted in a mounting medium. For immunofluorescence staining, tissue sections were blocked with 5% BSA and then incubated with PE-conjugated rat-anti-mouse CD8a (BioLegend). DAPI was used to stain nuclei. After washing, tissue sections were then mounted using ProLong Gold reagent (Invitrogen) and slides were digitalized using an All-in-One Fluorescence Microscope BZ-X700 and BZ-X Analyzer 1.4.0.1 (Keyence). To analyze tumor-associated leukocytes, excised tumors were minced with sterilized scissors and were dissociated using the GentleMACS Dissociator with Mouse Tumor Dissociation Kit (Miltenyi Biotech) according to the manufacturer's instructions with modifications. Briefly, after digestion with the provided enzyme mixture (45 min, 37 °C) and harvesting dissociated cells, the remaining tissues were further digested with 10% (v/v) trypsin containing 5 mM EDTA in Hank's balanced salt solution without calcium and magnesium for 15 min at 37 °C to improve recovery of macrophages and other myeloid leukocytes. The dissociated cells in suspension were then filtered through a 70 μm nylon strainer, treated with red blood cell lysis buffer and were counted by a cell counter. For flow cytometric analyses, cells were blocked with anti-FcR (clone 2.4G2; Bio X Cell) before incubating with a mix of fluorophore-conjugated antibodies (4 °C, 30 min) to detect different populations of tumor-associated leukocytes (gated as CD45$^+$) that included macrophages (gated as CD45$^+$CD11b$^+$F4/80$^{high}$Ly6C$^-$), monocytes (CD45$^+$CD11b$^+$F4/80$^+$ Ly6C$^{high}$), granulocytes/PMN (CD45$^+$CD11b$^+$ Ly6G$^{high}$), dendritic cells (DC, CD45$^+$CD11c$^+$F4/80$^-$), CD4 Th (CD45$^+$CD4$^+$), CD8 Tc (CD45$^+$CD8$^+$), NK (CD45$^+$NK1.1$^+$), and B cells (CD45$^+$CD19$^+$). Dead cells were excluded by 7AAD staining (Gating strategy also see Supplementary Fig. 7). Antibodies were from BioLegend, eBioscience, and BD Pharmingen. CD8 Tc that recognize the tumor-specific antigens p15E and ADPGK were detected by H-2Kb MuLV p15E Tetramer and H-2Db ADPGK Neoepitope Tetramer (both from MBL Life Science), respectively. Effector memory T cells within the CD8 Tc population were gated as CD44$^+$CD62L$^-$. Treg and Th1 cells within CD4 Th were gated as Foxp3$^+$ and IFNγ$^+$, respectively, after intracellular staining (Intracellular Cytokine Staining kit from BD Pharmingen was used for detecting IFNγ). CD8 Tc and NK cells expressing granzyme B (GranzB) were detected by intracellular staining with a mouse anti-human/mouse granzyme B antibody (BioLegend). To detect PMN expressing ROS, total dissociated cells were treated with 1 μM PMA (4 h, 37 °C) in the presence of oxidation-sensitive dye DCFDA (5 μM, Invitrogen) followed by washing and flow cytometry[48,49]. To detect macrophage expression of cytokines, total dissociated cells were treated with brefeldin A (BioLegend, 5 μg/ml) for 4 h at 37 °C and cell surface staining for cell type gating, followed by fixation and permeabilization and intracellular staining using cytokine-specific fluorophore-conjugated antibodies or biotin-conjugated primary antibodies in combination with streptavidin-PE. FACS data were collected from BD LRSFortessa (BD Bioscience) using BD FACSDiva 8.0.2, followed by analyses using FlowJo 10.6 (Tree Star). Details of antibodies used in this study are shown in Supplementary Table 1.

**NanoString gene expression analyses**. Tumors were resected from WT and Sirpα$^{-/-}$ mice 12 h and 18 h after 8 Gy IR (n = 2 mice per time point), followed by isolation of total RNA with TRIzol Reagent (The Invitrogen Life Technologies). RNA samples of the two-time points were pooled, of which 100 ng RNA was used for gene expression detection with the NanoString Mouse Immunology Panel. The nCounter XT protocol was used for hybridization and the data was collected by the Nanostring nCounter® SPRINT Profiler followed by analyses with nSolver 2.6 software (NanoString Technologies). Background hybridization by spiked-in negative controls was deducted. A normalization factor was calculated from the spiked-in exogenous positive controls in each sample and applied to the raw counts from nCounter™ output data. The complete gene expression profiling data have been deposited in the Gene Expression Omnibus (GEO) database under accession code GSE149882.

**Macrophage phagocytosis assay.** BMDMs from WT or Sirpα$^{-/-}$ mice cultured in 24-well plates were treated with LPS from Escherichia coli O111:B4 (20 ng/mL, Sigma), CpG 1826 (1 μg/mL, Invivogen), HMGB1 (100 ng/mL, Sigma), IL-1β (20 ng/ml, Peprotech), or IL-6 (20 ng/mL, Peprotech) for 18 h[50], followed by incubating with CFSE-labeled MC38 at a ratio of 3 MC38 cells for every 1 macrophage for 45 min. After washing, macrophage phagocytosis was analyzed microscopically or by FACS. To test phagocytosis towards irradiated cancer cells, CFSE-labeled MC38 cells were first treated with a fraction of X-ray radiation at a dosage of 8 Gy. Totally, 8–18 h after irradiation, conditioned medium from irradiated MC38 cells were collected and tested for its capacity to activate macrophages. Irradiated MC38 cells were added into cultures containing WT and Sirpα$^{-/-}$ BMDMs followed by co-incubation and assessment of phagocytosis. In some experiments, 20 μg/ml of rat-anti-mouse CD47 or soluble murine SIRPα extracellular domain fusion protein (mSirpα.ex) was added into WT BMDM cultures to block the CD47-SIRPα interaction. Phagocytic indexes were quantified by the number of macrophages that ingested at least one target per 100 macrophages analyzed.

**CD8 T cell cytotoxicity assay.** To isolate tumor-infiltrated Tc, tumor tissues were dissociated and cells were harvested. Cells were then allowed to adhere to a cell culture dish for 2 h followed by collecting non-adherent cells, a step that enriches TILs (generally comprising 30–40% Tc and 10–20% Th cells). Tc was then further isolated from TILs using the CD8$^+$ T Cell Isolation Kit (Miltenyi Biotech). Alternatively, Tc was expanded from TILs in vitro by anti-CD3/CD28 ligation or by tumor antigen-loaded Sirpα$^{-/-}$ macrophages. To test Tc cytotoxicity, cultured cancer cells ($5 \times 10^5$) were labeled with CFSE and co-incubated with freshly prepared Tc (>95% viability) of various numbers (tumor cell: T cell = 1:1–1:10) in the presence of PI. After incubation for 3, 6, or 24 h, Tc cytotoxicity-induced tumor cell death was examined by microscopy and the remaining live tumor cells (green) were counted, which was verified by flow cytometry to detect CFSE$^+$PI$^+$ cells after collecting all cells by trypsinization. Real-time videos recording Tc killing tumor cells were made using a Nikon camera (DS-Qi1MC) with NIS-Elements software BR 4.20.00 (Nikon Instruments) that captures images at 10-s intervals over a 3-h period.

**PMN and monocyte chemotaxis.** MC38 tumors without IR and three days post-IR were excised from WT and Sirpα$^{-/-}$ mice. After being weighed, tumors were minced followed by culturing in RPMI-1640 with 10% FBS. After 24 h, the culture supernatants (tumor-conditioned medium) were collected and tested for the presence of MCP-1 and KC chemokines by a multiplex assay (BioLegend), and the capability of driving monocyte/MDSC or PMN chemotaxis by in vitro chemotaxis assays[49,51]. Ly6C$^+$ monocytes/MDSCs and Ly6G$^+$ PMNs ($1 \times 10^6$ each) isolated from WT mouse bone marrow[52] were labeled with CFSE and placed into the upper chamber of the transwell device in a 24-well plate. The collected tumor-conditioned medium (0.5 ml) was added into the lower chamber followed by incubation at 37 °C for 2 h. Chemotaxis of monocytes/MDSC and Ly6G$^+$ PMN into the lower chamber was quantified by a SpectraMax iD5 Multi-Mode Microplate Reader with SoftMax Pro 7 Software (Molecular Devices) and calculated against the total cells loaded. Control experiments were performed using medium without culturing tumor tissues.

**Assaying myeloid leukocytes inhibiting T cells.** Intratumoral myeloid leukocytes were isolated from dissociated tumors using EasySep™ Mouse CD11b Positive Selection Kit (StemCell). To test their inhibition on T cells, myeloid leukocytes were added into the anti-CD3/CD28-induced T cell proliferation system at a ratio of CD11b$^+$ cells: splenocytes = 1:5 followed by coculturing for 4 days. The proliferation of T-cells cocultured with intratumoral myeloid leukocytes isolated from WT or Sirpα$^{-/-}$ tumor-bearing mice were then evaluated microscopically and by FACS that determined CFSE dilution[49].

**In vitro expansion of TILs by anti-CD3/CD28.** Enriched TIL fractions (~50% T cells) were induced to proliferate in a 24-well plate ($1 \times 10^6$/well) that had been immobilized with anti-CD3 antibody (1 μg/mL) in the presence of soluble anti-CD28 antibody (0.5 μg/mL, both rat anti-mouse, BioLegend) in RPMI-1640 medium containing 10% FBS, 1% penicillin/streptomycin, 2 mM L-glutamine, 50 μM β-mercaptoethanol. After 24 h (day 2), the culture was supplemented with 50 IU/ml recombinant IL-2 (rIL2, Hoffmann La Roche, kindly provided by the National Cancer Institute Biological Resources Branch Preclinical Repository, Rockville, MD). To accommodate the rapid T cell proliferation, cells were replenished with a new medium containing IL-2 every three days and the cell density was maintained at $1 \times 10^6$ cells/ml. Cells were harvested on day 10 since T cells often had proliferated more than eight generations and had expanded over 100-fold. These expanded T cells, of which 70–80% were Tc, were termed TIL-MC38 or TIL-KPC dependent on the tumor from which the initial TILs were isolated, and these expanded TILs were further tested for tumor cytotoxicity in vitro and in vivo.

**Ex vivo and in vitro expansion of tumor-specific Tc by tumor antigen-presenting Sirpα$^{-/-}$ macrophages.** For ex vivo expansion of tumor-specific Tc,

tumors in WT or Sirpα$^{-/-}$ mice were excised without TDLN 15 min post 8 Gy. Tumor tissues were then dissociated into individual cells using the Mouse Tumor Dissociation Kit followed by further digestion with 10% (v/v) trypsin containing 5 mM EDTA in Hank's balanced salt solution. Dissociated tumor tissue cells were cultured in RPMI-1640 containing 10% FBS for ex vivo expansion of tumor-specific Tc. To ensure macrophage and T cell contact, dissociated tumor tissue cells were cultured at a high density, with $2 \times 10^6$ tumor tissue cells in 200 μl RPMI-1640 containing 10% FBS. The cells were added in the upper chamber of a 24-well transwell device with filters (0.4 μm diameter pore) with the lower chamber filled with 2 ml RPMI1640 plus 10% FBS to provide nutrients. Cell samples were collected at various time points and the frequency of Tc therein was assessed by flow cytometry. Irradiated WT tumors were also infused with Sirpα$^{-/-}$ BMDM ($1 \times 10^4$ BMDM/mm$^3$ tumor) prior to tissue dissociation. For in vitro expansion of tumor-specific Tc, Sirpα$^{-/-}$ BMDMs ($2 \times 10^5$/well, 24-well plate) produced by M-CSF from bone marrow cells were incubated with tumor-dissociated cells and debris prepared from 8 Gy-irradiated tumors. After overnight incubation (18 h, 37 °C), a time period allowing Sirpα$^{-/-}$ BMDMs to phagocytose and process tumor antigens, the Sirpα$^{-/-}$ BMDM-containing wells were gently washed, followed by adding enriched TIL ($1 \times 10^6$/well; TIL: BMDM of 5:1). The Sirpα$^{-/-}$ BMDM-TIL co-culture was then maintained (37 °C, 5% CO$_2$) for 10 days in RPMI-1640 medium containing 10% FBS, 1% penicillin/streptomycin, 2 mM L-glutamine and 50 μM β-mercaptoethanol, with 50 IU/ml recombinant IL-2 added on day 2. The culture was inspected daily by microscopy and T cell proliferation was generally apparent on day 3. Cell samples were also subjected to flow cytometric analysis of contents of Tc and Th cells. To accommodate rapid T cell proliferation, the IL-2-containing medium was replenished every 3 days (adding a new medium or changing the medium) and the cell density was maintained below $1 \times 10^6$ cells/ml. After 10 days, the T-cell number had generally increased 20-fold, from the $1 \times 10^6$ to approximately $2 \times 10^7$ T cells of which >95% were Tc; thus, these expanded cells were termed Tc-MC38 or Tc-KPC, and were tested for cytotoxicity against respective tumor cells in vitro and in vivo by adoptive T-cell therapy.

**Adoptive T-cell therapy.** In vitro expanded T cells, either by anti-CD3/anti-CD28 or by tumor antigen-presenting Sirpα$^{-/-}$ macrophages, were used to treat WT mice bearing the same type of tumor (MC38 or KPC). Once palpable tumors grew >200 mm$^3$ or were collective >600 mm$^3$, the tumor-bearing WT mice were i.v. administered $0.5–2.0 \times 10^7$ T cells, along with a dose of IL-2 (25,000IU, i.p.) that was additionally given for the next 5 consecutive days[53,54]. Subsequent rounds of T-cell infusion were performed at 3-day intervals (MC38) or for 4 consecutive days in total (KPC); mice with a single or multiple tumor(s) received two or four rounds of T-cell infusion, respectively.

**Statistical analysis.** Data are presented as mean ± SEM and/or as individual values. All graphs and statistical analyses were generated and performed using Prism 7.0 (GraphPad Software). Statistical significance was calculated using the Student's t test ($k = 2$) or one-way analysis of variance ($k > 2$). For post hoc analyses of ANOVAs, Tukey's (one-way) was used to determine statistical significance among multiple comparisons, with an experiment-wise error rate of 0.05. For analysis of overall survival (Kaplan–Meier curves), Mantel–Cox log-rank test was used to determine statistical significance. Data were considered statistically significant when $P < 0.05$.

**Reporting summary.** Further information on research design is available in the Nature Research Reporting Summary linked to this article.

## Data availability
NanoString gene expression profile has been deposited in the Gene Expression Omnibus (GEO) database under accession code GSE149882. Source data are available as a Source Data file. The remaining data are available within the Article, Supplementary Information or available from the authors upon request. Source data are provided with this paper.

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

## Acknowledgements

The authors thank the Georgia State University Animal Resources Program for assisting many experiments; This work was supported, in part, by grants from the National Institutes of Health (CA241271 and AI106839), a Research Scholar Grant (RSG-15-182-01) from the American Cancer Society (C.B.), a Georgia Research Alliance (GRA) Venture Development grant, a Biolocity Innovation & Commercialization grant, a Careers in Immunology fellowship from American Association of Immunologist (Z.B.), a Molecular Basis of Disease fellowship from Georgia State University (K.K.) and an Ahmed T. Abdelaal Molecular Genetics and Biotechnology fellowship from Georgia State University (K.K.).

## Author contributions

Y.L., C.B., and Z.B. conceived of the study. Z.B. and Y.L. designed all the experiments. Z.B., L.S., and K.K. performed the experiments, analyzed and interpreted the data, and the statistical analyses. C.B. and K.Z. provided technical advice and data interpretation. Z.B, K.K., K.Z., and Y.L. wrote the paper.

## Competing interests

Z.B., L.S., K.K., K.Z. and Y.L. are the inventors of IP related to SIRPα-deficient-based macrophage and tumor-specific T-cell therapies; Y.L. is a co-founder, scientific advisor and shareholder in SIRPant Immunotherapeutics, Inc. C.B has no competing interests.
