## [Peer Review File · Nature Communications]

REVIEWER COMMENTS

Reviewer #2 (Remarks to the Author): with expertise in tumor associated macrophages and radio/immunotherapy

This manuscript describes a study showing a pronounced enhancement of anti tumour immunity after ionizing radiation in Sirpa deficient mice compared to WT. They show that this is due to Sirpa deficient macrophages having a better capacity for antigen presentation. These are interesting results but the manuscript currently has a number of deficiencies. First the results are essentially all with one cell line, MC38. Second a number of the experiments do not have controls and most of those using KPC are underpowered. Third the macrophages in general from Sirpa^{-/-} mice are not characterized, only TAMs.

Figure 1 shows growth curves from MC38, PAN02 and KPC tumors s.q. with and without radiation in WT mice and in Sirpa^{-/-} mice showing marked tumor regression after XRT. These curves however are shown only after XRT when the tumors had reached approx. 150mm³ in volume. What effect was there on tumor growth itself in Sirpa^{-/-} mice? Also this figure shows the growth for a few mice after being given anti PD-L1 antibodies. I don't think these add to the study as there are only a few and there is no followup on this point.

Figure 2 then asks whether irradiation of a primary s.q. tumor will also influence regression by a second, ie an abscopal effect. B shows that there is an abscopal effect that is revealed if the abscopal tumor is less than 100 mm³ for MC38. The data is not presented so that any kind of quantitative comparison can be made- for example Kaplan-Meier statistics, or sizes at given times etc. The data for KPC in c is only presented as images of mice without any measurements. There are only 3 control mice, 3 abscopal mice and 4 peritoneal abscopal. This is certainly not sufficient. Two mice are given aPD-L1- again this is not relevant.

They show that the mice that survive are resistant to rechallenge. They also indicate that the mice that survive carry antibodies to MC38. This is not pursued further. Do these antibodies develop after XRT alone? Tumor formation? They show that passive transfer of T cells from the spleens of tumor free mice confer rejection. What they make of the the antibodies is unclear. B cells are not followed for the most part.

This brings up the point that they refer to MC38 as poorly immunogenic. However most workers regard it as at least moderately immunogenic. This is probably because it is mismatch repair deficient and generates substantial numbers of mutations.

Because Sirpa is mainly expressed in myeloid cells they now ask whether the Sirpa deficiency affects the immune response due to altered macrophages. First they show that depletion of macrophages in Sirpa^{-/-} mice reduces the antitumor effect after radiation. Somewhere they should indicate the extent of depletion that they found with the antibodies given.

They also ask what affect introducing Sirpa^{-/-} macrophages could have. They do this by generating BMDM Sirpa^{-/-} mice and then injecting these BMDM either iv or into the tumor. (No detailed description of the BMDM differentiation in tissue culture is provided). No profiling of these the endogenous Sirpa^{-/-} macrophages or of the BMDM is provided. WT BMDM are not used as a control. In Fig 3d, what 1x or 2x means is not obvious. Does 2 rounds mean 2 macrophage applications as well? I am surprised by the large number of BMDM injected IV. The Kaplan Meier plot should have statistics with it. Again KPC is not shown quantitatively and only 3 mice in each group and without controls of transferring WT macrophages or transferring +/+ into WT mice etc.

Figure 4 then characterizes the macrophages within the tumors. For this reader this figure fails to characterize the macrophages in Sirpa^{-/-} mice for antigen presenting molecules for example. TAMs are also not characterized. Instead the figures examine the properties of infused BMDM. TAMs should be examined and it would be interesting to examine the abscopal tumors after the primary has had XRT. Given the literature (ad my own personal experience) I am surprised that in Fig. 4c that there was no increase in CD11b⁺ cells after XRT. I dont fully understand the labels on f and g. The x axis is labelled as increased F4/80 but then the y axis would be MHC I etc. Yet it would seem to be the reverse. Overall there should be information on markers on Sirpa^{-/-}

macrophages compared to WT, both naïve and TAMs. It also might be interesting to see the polarization markers that have been associated with radiation in the literature. Also in the WT is Sirpa expression heterogeneous and does it inversely correlate with the antigen presentation markers?

Figure 5 examines the T cell response based upon Sirpa genotype. This figure shows in h that CD8 T cell depletion reduces the regression due to XRT and Sirpa deficiency.

Figure 6 looks in more detail at the infiltrating host immune cells in the Sirpa^{-/-} tumors compared to those in WT before and after XRT.

Figure 7 now addresses the hypothesis that Sirpa^{-/-} macrophages are effective antigen presenting cells. (I didn't readily find how soon after XRT the macrophages were injected.) Panel c lacks a size bar. This figure shows that coculture of Sirpa^{-/-} BMDM with irradiated MC38 cells and tumor derived T cells leads to proliferation of CD8 but not CD4 T cells. Again controls with WT BMDM could be provided. G shows experiments with only 3 points.

Nonetheless this experiment makes it clear that T cells active against the tumor cells have been generated.

Throughout numbers of mice- n- in each group should be given. They are in some cases, but not in others Fig 3f for example.

No animal ethics information is provided.

Discussion indicates that "this study unveils an important mechanism by which malignant tumors resist RT..." This is not really true. What it reveals is that in the absence of Sirpa that immune response is enhanced. This is an aberrant situation and does not provide evidence for this general statement." They also say that "SIRPa deficient intratumoral macrophages respond to RT by manifesting a superior anti tumor capacity." They do not show that radiation has any effect on the macrophages or that they even need to be in the TME to have this effect.

Extended Figure 9 introduces two new cell lines and a completely different experiment. It should be removed or else requires more thorough exploration.

Reviewer #3 (Remarks to the Author): with expertise in radio/immunotherapy

The aim of this study was to explain how SIRP^{alpha}^{-/-} status had improved RT efficacy on three preclinical cancer models. The authors demonstrated that SIRP, expressed by macrophages, regulated antigen presentation and tumor specific cytotoxic T cells radio-induced activation. This is a very interesting work, especially in this current era whereby radio-induced immune response is currently described to be depend on APC activation followed by CD8⁺ activation tumor antigen specific. This study highlight an important role of macrophages in this mechanism. Authors detailed each step of experiments to facilitate the understanding of their objectives for uninitiated readers.

The conclusions and perspectives of this study will allow to develop new therapeutic to improve radiation therapy efficacy.

Nonetheless while interesting, there are some major and minor limitations of the current data.

1. The Figure 1 is clear. To improve it I suggest adding the RT dose used in the Fig1b.

The author have evaluated effect of different RT doses delivered using one or several fractions. I'm not sure that the Fig1e presents a real interest here.

However it's unexpected that RT present no effect for WT model whatever the dose! Concerning MC38 and KPC, I invite the authors to insert in their references Jones et al., EMBO Molecular Medicine 2018 (10:e9342) and to discuss these results please! Moreover, Jones et al have also observed a tumor growth delay after 1x10Gy which was improved after RT + anti-PDL1.

Concerning the suppl Fig 1, the authors used two dose rates to deliver 20Gy. For SIRPa^{-/-}, they

highlighted a better secretion of IL6, TNF α , IL12 and IL1 β after 20Gy delivered using 2Gy/min to compare to 1.2Gy/min. It's an interesting result which needs to be discuss!

2. The figure 2 investigated the radio-induced abscopal effect for SIRPa $^{-/-}$ model. Interestingly author demonstrated this effect using a standard contralateral tumor injection and also using un-irradiated intra peritoneal lesion. In the Fig2b experiments, authors observed an effect of initial tumor volume on response. As previously this result was not explained and discussed in the manuscript. It could be interesting to compare TME for tumor <100mm 3 and >200mm 3 which could provide some answers.

3. The figures 3 clearly demonstrated that the RT efficacy for SIRPa $^{-/-}$ model was dependent on macrophages. All experiments were well explained except the last graph of the Fig3d. The reader will need some explanations in the legend for this experiment.

4. The figure 4 demonstrated that the SIRPa $^{+/+}$ model lacked the capacity for immunogenic antigen presentation.

To improve the understanding of the fig4a author could insert CD45 in x axis legend.

In the sup fig 6 authors details their gating process to characterize each immune cells. Please explain how you can highlight a radio-induced decrease in macrophage in fig 4b whereas they don't observe significant modification of myeloid cell quantity (which include macrophages!) in figure 4c (for SIRPa $^{-/-}$ model). Could you please explain and discuss these results?

5. Finally, authors have to investigate if the phagocytosis capacity of macrophages is modified by SIRP depletion. There are some explorations about this in supplemental data which need to be explain and discuss in the manuscript

Reviewer #4 (Remarks to the Author): with expertise in tumor associated macrophages (to replace Reviewer #1)

Bian and colleagues showed that SIRPa-deficiency elicited potent adaptive immune response following local radiation in murine syngeneic tumor xenograft models. After initially phenotypically demonstrating complete tumor xenograft rejection in SIRPa $^{-/-}$ deficient mice receiving local radiation treatment, the authors investigated the establishment of anti-tumor adaptive immune responses and pro-inflammatory remodeling of the tumor microenvironment. The authors suggested the potent anti-tumor adaptive immune responses were orchestrated via antigen-capture by SIRPa $^{-/-}$ macrophages and subsequently cross-priming and activated CTLs. Most experiments were well designed and executed with necessary controls included.

Major points:

1) Very interestingly, the authors suggested CD47 blockaded by intratumoral delivery of either functional antibody or soluble murine SIRPa ecto domain fell short to recapitulate the results observed in SIRPa $^{-/-}$ mice. However, this conclusion would be further strengthened using CD47 KO tumor xenograft inoculated in WT mice in addition to pharmacological blockade as shown in this manuscript, to compare whether CD47 KO tumor xenograft inoculated in WT mice achieves the same tumor rejection effect as observed in WT tumor xenograft in SIRPa $^{-/-}$ mice.

2) Data representation is a major obstacle hindering readers to analysis and comprehend. While the authors tried to present data in a more comprehensive way, due to the limited space of a figure yet too many panels presented, it's challenging to read in detail for almost all figures. The conclusion was strengthened after validated in multiples models, as the authors did with MC38, KPC and Pan02 lines. However, KPC and Pan02 models provided minor additional information that was absent from MC38 model. An alternative way would be, for example, to move figures related to KPC and Pan02 into supplementary information while presenting only MC38 related data in the main Figures.

3) In the "macrophages present antigens to activate tumor-specific Tc in situ" section, the conclusion of the first paragraph "These results suggest that ICD and DAMPs-activated Sirpa $^{-/-}$ macrophages drive Tc expansion in situ independent of peripheral lymphoid organs" was

insufficiently supported based on the results demonstrated in fig.7a. In this experiment, the authors failed to demonstrate neither tumor cell apoptosis nor the release of DAMPs and specifically which DAMPs were released and thus macrophages sensed and activated.

4) Supplementary Fig.6, gating strategy. The authors failed to gate singlet prior to analysis. Without singlet gating, all downstream analyses were rendered problematic, especially for xenograft tumor tissue as they are notoriously sticky even after thorough digestion and passed through strainer. Failure to exclude doublets could introduce false positive in flow cytometry analysis. The authors should re-analyze flow cytometry data after making sure singlet gating is applied.

Minor points:

Text related

1) Sentence "Despite the absence of the TDLN, irradiated tumor explants from Sirpa^{-/-} mice exhibited robust expansion of Tc similar to that which occurred in vivo." Please add indication to which experiment in figures or citation to which study did the "that which occurred in vivo" refers to.

2) Please provide detailed experiment method description regarding the ex vivo culture of tumor tissue experiment in fig.7a.

3) Data transparency. In Figure legend of Figure 1. Please indicate exactly how many experiments were repeated, instead of "at least three independent experiments"

Figure related

- Fig.4a, annotations of X- and Y- axis are missing.
- Fig.4c, statistic P value missing for WT panel.
- Fig.4h, authors should demonstrate transcriptome profiling heatmap of each biological replicates separately (n=2-4 as described in figure legend) instead of the representative sample or group average as shown.

Point-to-point response to reviewers

Reviewer #2

1. This manuscript describes a study showing a pronounced enhancement of anti tumour immunity after ionizing radiation in Sirpa deficient mice compared to WT. They show that this is due to Sipra deficient macrophages having a better capacity for antigen presentation. These are interesting results but the manuscript currently has a number of deficiencies. First the results are essentially all with one cell line, MC38. Second a number of the experiments do not have controls and most of those using KPC are underpowered. Third the macrophages in general from Sirpa^{-/-} mice are not characterized, only TAMs.

Response: We appreciate the reviewer's positive evaluation of our work. To address the issues raised by the reviewer, we have performed several additional experiments and thoroughly revised our manuscript. First, during the past five years we have tested all syngeneic cancer cell lines available to us to grow tumors in SIRP α ^{-/-} mice and subsequently observe their responses to RT or other preclinical cancer treatments. These cancer cell lines, all of which have a C57BL6/J genetic background, included EL4 lymphoma, colorectal carcinoma MC38 with or without CD47 expression, pancreatic ductal adenocarcinomas KPC, Pan02 and MT5, Lewis lung carcinoma LLC, and B16-F10 melanoma. In addition, these extensive experiments employed numerous different RT dosage schemes (4-20Gy, administered as either single or multiple fractions, etc.), and in some cases combined it with other synergistic treatments, e.g. anti-PD-1/L1 antibodies. However, given that the vast amount of our amassed data cannot feasibly fit into one manuscript, we mainly presented the data, namely MC38, KPC and Pan02 studies, most befitting to the field studying preclinical RT. Our hope is that these data will reveal the pronounced effects that SIRP α -deficient macrophages beneficially provide to RT-based treatment modalities against cancer. Second, in this revision, we have strengthened the presentation of our data, especially that which demonstrates the abscopal effect on KPC tumors (new Fig. 7, new Extended Data, Fig. 10). Furthermore, we have presented the control data in each figure and clarified the number of animals used for each panel of data. Third, as originally done with TAM, bone marrow-derived macrophages (BMDM) from SIRP α ^{-/-} mice are now characterized as well (new Extended Data, Fig. 4).

2. Figure 1 shows growth curves from MC38, PAN02 and KPC tumors s.q. with and without radiation in WT mice and in Sirpa^{-/-} mice showing marked tumor regression after XRT. These curves however are shown only after XRT when the tumors had reached approx. 150mm³ in volume. What effect was there on tumor growth itself in Sirpa^{-/-} mice? Also this figure shows the growth for a few mice after being given anti PD-L1 antibodies. I don't think these add to the study as there are only a few and there is no follow up on this point.

Response: We thank the reviewer for pointing out this important issue. Indeed, as suggested by others and our own studies, Sirp α ^{-/-} mice do have an inherent capacity to resist engraftment of cancer cells and nascent tumor formation, likely due to Sirp α ^{-/-} macrophages becoming activated during tumor engraftment – which in itself functions as a vaccination – and subsequently phagocytosing the injected

cancer cells. Prior to studying the effect of RT on tumor growth in Sirp $\alpha^{-/-}$ and WT mice, we carefully monitored tumor formation following engraftment in Sirp $\alpha^{-/-}$ and WT mice. As shown in new Fig. 1a, MC38 cells were subcutaneously (s.c.) engrafted in Sirp $\alpha^{-/-}$ and WT mice at 5×10^3 , 5×10^4 , 5×10^5 or 2×10^6 cells per mouse. As expected, when fewer cancer cells were engrafted (5×10^3 cells/mouse), Sirp $\alpha^{-/-}$ mice exhibited significantly slower MC38 tumor growth compared to WT mice, suggesting that Sirp $\alpha^{-/-}$ mice have pre-existing immune control – to a certain extent – against syngeneic MC38 cells. However, when the number of engrafted cells was increased (5×10^4 , 5×10^5 or 2×10^6 cells/mouse), MC38 tumors formed and grew at a similar rate in Sirp $\alpha^{-/-}$ and WT mice. Pan02 and KPC cancer cells exhibited a similar phenomenon. Therefore, to explore the effect of RT on syngeneic tumors in both Sirp $\alpha^{-/-}$ and WT mice, we established tumors by engrafting a higher concentration of cancer cells (5×10^5 per mouse). According to the reviewer's suggestion, we have shown the tumor growth curves before and after RT (new Fig. 1d). We also agree with the reviewer that the number of mice subjected to PD-L1 blockade were too few and thus have removed these results from the revision.

3. Figure 2 then asks whether irradiation of a primary s.q. tumor will also influence regression by a second, ie an abscopal effect. B shows that there is an abscopal effect that is revealed if the abscopal tumor is less than 100 mm³ for MC38. The data is not presented so that any kind of quantitative comparison can be made- for example Kaplan-Meier statistics, or sizes at given times etc. The data for KPC in c is only presented as images of mice without any measurements. There are only 3 control mice, 3 abscopal mice and 4 peritoneal abscopal. This is certainly not sufficient. Two mice are given aPD-L1- again this is not relevant.

Response: First, we apologize for not presenting the data in a manner conducive to quantitative comparisons. In the revision, we performed quantitative analysis (Kaplan-Meier statistics – Mantel-Cox Log-Rank Test) for all tumor (primary and abscopal) growth, as the reviewer suggested. All the revised figures now include Kaplan-Meier survival curves and appropriate statistical analysis. Second, we have extensively tested RT-induced abscopal effects exhibited in Sirp $\alpha^{-/-}$ mice in several tumor models. However, to avoid shoehorning the vast amount data, we only present the MC38 and KPC studies in this manuscript. We now present additional whole-body images of KPC-bearing Sirp $\alpha^{-/-}$ mice demonstrating abscopal effects in the new Fig. 7d and incorporated even more images of KPC-bearing Sirp $\alpha^{-/-}$ mice, as well as WT mice, in new Extended Data Fig. 10. As the abscopal effect is dependent upon activation of anti-tumor Tc, we have moved the abscopal effect results from Fig. 2 to Fig. 7, following the analysis of Tc activation by tumor antigen-presenting Sirp $\alpha^{-/-}$ macrophages. Finally, we removed the results of anti-PD-L1 from the new Fig. 2 accordingly.

4. They show that the mice that survive are resistant to re-challenge. They also indicate that the mice that survive carry antibodies to MC38. This is not pursued further. Do these antibodies develop after XRT alone? Tumor formation? They show that passive transfer of T cells from the spleens of tumor free mice confer rejection. What they make of the antibodies is unclear. B cells are not followed for the most part. This brings up the point that they refer to MC38 as poorly immunogenic. However most workers

regard it as at least moderately immunogenic. This is probably because it is mismatch repair deficient and generates substantial numbers of mutations.

Response: Indeed, mice that had survived had produced antibodies against the cancer cells that had been engrafted, and we believe that this occurrence is an important part of long-term anti-tumor immunity in these mice. These antibodies were only developed in $\text{Sirp}\alpha^{-/-}$ mice that had survived tumor engraftment and treatment. In other words, tumor-specific antibodies were generated in $\text{Sirp}\alpha^{-/-}$ mice only after tumors had formed and were subsequently eliminated by XRT. XRT alone in cancer-naïve mice (i.e., mice that were not challenged with tumor engraftment) did not produce antibodies against the cancer cells. This is shown more clearly in new Fig. 8c, in which we screened all mouse sera for their capacity to label cancer cells in vitro, and we found that only the sera from tumor-eradicated $\text{Sirp}\alpha^{-/-}$ mice comprised cancer-specific antibodies that labeled the surface of cancer cells.

Aside from demonstrating the specificity of these anti-cancer sera from tumor-eradicated $\text{Sirp}\alpha^{-/-}$ mice, we also analyzed their capacity to exert cancer-killing activities. As shown in new Fig. 8, these anti-cancer sera mediated robust cancer cell death in vitro through complement-dependent cytotoxicity (CDC) (Fig. 8d, left) and Fc-mediated phagocytosis (Fig. 8d, right). Preemptively vaccinating WT mice with this anti-cancer sera (specific for MC38) from tumor-eradicated $\text{Sirp}\alpha^{-/-}$ mice prevented MC38 tumor formation following engraftment (Fig. 8f), suggesting that this anti-MC38 sera is sufficient to prevent the formation of nascent MC38 tumors. This observation corroborates the notion that production of anti-cancer sera against tumors in $\text{Sirp}\alpha^{-/-}$ mice is an essential component that prevents tumor recurrence. However, the anti-cancer sera has a limited capacity to control the growth of established tumors. When MC38 tumors were fully established in WT mice, anti-cancer sera treatment largely failed to regress tumors and only moderately delayed tumor growth (Fig. 8f). Given that the present study emphasizes the elimination of syngeneic tumors, we did not further pursue the role of B cells, instead, we focused on $\text{Sirp}\alpha^{-/-}$ macrophages capacity to recall tumor-specific T cells, which exhibited a potent capacity to eradicate large tumors (100-400mm³). Finally, we agree with the reviewer that MC38 is not poorly immunogenic per se given that it comprises a substantial number of mutations, toward which an immune response may be generated and equally suppressed. To correct this, we removed 'poorly immunogenic' in direct reference to MC38 in the revision.

5. Because *Sirpa* is mainly expressed in myeloid cells they now ask whether the *Sirpa* deficiency affects the immune response due to altered macrophages. First they show that depletion of macrophages in *Sirpa*^{-/-} mice reduces the antitumor effect after radiation. Somewhere they should indicate the extent of depletion that they found with the antibodies given. They also ask what affect introducing *Sirpa*^{-/-} macrophages could have. They do this by generating BMDM *Sirpa*^{-/-} mice and then injecting these BMDM either iv or into the tumor. (No detailed description of the BMDM differentiation in tissue culture is provided). No profiling of these the endogenous *Sirpa*^{-/-} macrophages or of the BMDM is provided. WT BMDM are not used as a control. In Fig 3d, what 1x or 2x means is not obvious. Does 2 rounds mean 2 macrophage applications as well? I am surprised by the large number of BMDM injected IV. The Kaplan Meier plot should have statistics with it. Again KPC is not shown quantitatively and

only 3 mice in each group and without controls of transferring WT macrophages or transferring +/- into WT mice etc.

Response: We appreciate the reviewer's constructive comments. First, we added a new panel to show the extent of macrophage depletion by two methods in new Fig. 2 (original Fig. 3). Flow cytometry analysis (Fig. 2b) showed that Cl2MDP and α CSF1R treatment depleted about 70-80% intratumoral macrophages, which are in line with previous reports (Jones et al. *EMBO Mol Med.* 2018 Dec; 10(12): e9342. new Ref #21; Wang et al. *BMC Immunology* volume 12, 2011). Second, we provided a detailed description of the procedure by which we differentiated BMDMs in culture in the revised Method section. Third, according to the reviewer's comment, we provided profiling of the endogenous Sirp $\alpha^{-/-}$ BMDM, as well as WT BMDM (served as control) in new Extended Data, Fig. 4. In all figures, 1x or 2x means one or two rounds of the treatment arm (macrophage infusion plus IR, or antibody injection plus IR). For example, in Fig. 3d (now Fig. 2e), 2 rounds equates to treating mice twice with *i.t.* infusion of Sirp $\alpha^{-/-}$ BMDM and twice with IR, with both modalities administered on the same day (i.e., *i.t.* BMDM + IR on Day 12, followed by *i.t.* BMDM + IR on Day 15). As for the large number of BMDM used in *i.v.* injection, this high concentration of BMDM was based upon our observations of how many *i.v.* injected BMDM can be recruited into the tumor. As shown in new Fig. 2f, tracing IV injected fluorescent BMDM showed that $\sim 10^7$ BMDM needed to be IV injected to comprise $\sim 40\%$ of the total intratumoral macrophages. Finally, according to reviewer's comment, Kaplan-Meier survival curves and appropriate statistical analysis (Mantel-Cox Log-Rank Test) have been provided for all tumor-bearing mice. Based on the new results, we have also revised the text accordingly (Page 7-8, revisions are in red).

6. Figure 4 then characterizes the macrophages within the tumors. For this reader this figure fails to characterize the macrophages in Sirp $\alpha^{-/-}$ mice for antigen presenting molecules for example. TAMs are also not characterized. Instead the figures examine the properties of infused BMDM. TAMs should be examined and it would be interesting to examine the abscopal tumors after the primary has had XRT. Given the literature (ad my own personal experience) I am surprised that in Fig. 4c that there was no increase in CD11b⁺ cells after XRT. I don't fully understand the labels on f and g. The x axis is labelled as increased F4/80 but then the y axis would be MHCI, etc. Yet it would seem to be the reverse. Overall there should be information on markers on Sirp $\alpha^{-/-}$ macrophages compared to WT, both naïve and TAMs. It also might be interesting to see the polarization markers that have been associated with radiation in the literature. Also in the WT is Sirp α expression heterogeneous and does it inversely correlate with the antigen presentation markers?

Response: In new Fig. 3 (original Fig. 4), we have characterized both the TAM (Fig. 3c) and infused BMDM macrophages in Sirp $\alpha^{-/-}$ mice and WT mice for antigen presenting molecules, such as MHC-I/II and co-stimulatory factors CD80, CD86, etc., as well as macrophage polarization markers iNOS and CD206 (Fig. 3, c and h). We have also characterized the endogenous BMDM in both Sirp $\alpha^{-/-}$ and WT mice (new Extended Data Fig. 4), and the results show that Sirp $\alpha^{-/-}$ BMDM express higher levels of MHC-I/II, co-stimulatory factors CD80 and CD86, and M1-marker iNOS but lower level of M2 marker CD206 than WT BMDM. We agree with the reviewer that examining the abscopal tumor TME after the

primary has had XRT would provide insight into how the abscopal effect proceeds. We have thus analyzed the abscopal tumor (at different stages: $<100\text{mm}^3$ and $>200\text{mm}^3$) after the primary tumor has had XRT. As shown in new Extended Data, Fig. 11a, there was significantly more infiltration of CD8 Tc into small ($<100\text{mm}^3$) non-irradiated tumors than large ($>200\text{mm}^3$) non-irradiated tumors, supporting the role of Tc in abscopal control of smaller tumors (Extended Data Fig. 11a). In contrast, the TME of large non-irradiated tumors was still dominated by pro-tumor myeloid cells. We further analyzed the TAMs in the large non-irradiated tumor that were not eliminated by the abscopal effect. As shown in new Extended Data Fig. 11b, unlike TAMs in irradiated tumors, which displayed significantly increased antigen presentation capacity and M1-phenotypic activation, TAMs in non-irradiated tumors retained a lower expression of antigen presentation machinery MHC-I/II, CD80/86, OX40L, M1-phenotypic marker iNOS but a higher level of M2-phenotypic marker CD206.

We apologize for the improper labeling of figures in original Fig. 4c and have corrected it in new Fig. 3b. There was, indeed, a significant increase in CD11b⁺ cells after XRT. However, the appearance of this increase in CD11b⁺ cells was inadvertently masked in the original Fig. 4 because we showed it on a scale warped by the high number of T cells, which were at least six-fold greater in quantity than CD11b⁺ cells. In new Fig. 3a-b, we separately demonstrated the fluctuations in T cell and CD11b⁺ cell populations, and as can be seen, the number of CD11b⁺ cells significantly increased following XRT. We have also fixed the labels on old Fig. 3 f and g (new Fig. 3, c and h). According to the reviewer's suggestion, we also tested two polarization markers, iNOS (M1) and CD206 (M2) on TAM from both Sirp $\alpha^{-/-}$ and WT mice before and after XRT (new Fig. 3, c and h). Indeed, our results indicated that endogenous Sirp $\alpha^{-/-}$ macrophages displayed more of an M1 phenotype than endogenous WT macrophages, and tumor localized radiation further activated the infused Sirp $\alpha^{-/-}$ macrophages toward M1 polarization. Finally, we tested whether Sirp α on WT macrophages is inversely correlated with the antigen presentation markers. As shown in the figure below, flow cytometry data suggest that Sirp α expression on WT mouse BMDMs likely is homogeneous. When WT BMDMs were incubated with tumor cell culture medium (tumor conditioned medium), the Sirp α expression was increased, whereas the surface levels of MHC-I/MHC-II and co-stimulators CD80 and CD86 were decreased. In contrast, when WT BMDMs were treated with LPS, Sirp α expression on WT BMDMs was decreased (which is in agreement with previous studies by us (Zhu et al. *J Allergy Clin Immunol.* 2013, 132:426-36.e8) and others (Kong et al. *JEM.* 2007, 204:2719-31), whereas the surface levels of MHC-I/MHC-II, CD80 and CD86 were increased. These results collectively suggest that macrophage Sirp α is inversely correlated with the antigen presentation markers.

Reviewer only Figure 1. Macrophage SIRP α inversely correlates with the antigen presentation markers. **a**, SIRP α surface label in WT macrophages. **b**, Macrophage SIRP α is upregulated by tumor conditioned medium but downregulated by LPS treatment. **c**, Macrophage SIRP α inversely correlates with the antigen presentation markers

Figure 5 examines the T cell response based upon Sirpa genotype. This figure shows in h that CD8 T cell depletion reduces the regression due to XRT and Sirpa deficiency.

Response: Indeed, Figure 5 (now Figure 4) demonstrates that the activation of tumor-specific Tc is the key for eliminating tumor by activated Sirpa $^{-/-}$ macrophages.

8. Figure 6 looks in more detail at the infiltrating host immune cells in the Sirpa $^{-/-}$ tumors compared to those in WT before and after XRT.

Response: Indeed, through analyzing the tumor microenvironment including various infiltrating immune cells in Sirpa $^{-/-}$ tumors compared to those in WT before and after XRT, Figure 6 (now Figure 5) demonstrates the reprogramming toward an anti-tumor TME by Sirpa $^{-/-}$ macrophages following XRT.

9. Figure 7 now addresses the hypothesis that Sirpa $^{-/-}$ macrophages are effective antigen presenting cells. (I didn't readily find how soon after XRT the macrophages were injected.) Panel c lacks a size bar. This figure shows that co-culture of Sirpa $^{-/-}$ BMDM with irradiated MC38 cells and tumor derived T cells leads to proliferation of CD8 but not CD4 T cells. Again controls with WT BMDM could be provided. G shows experiments with only 3 points. Nonetheless this experiment makes it clear that T cells active against the tumor cells have been generated.

Response: For adoptive transfer (*i.t.* or *i.v.*) of macrophages, Sirp $\alpha^{-/-}$ and WT macrophages were injected into WT mice immediately after administering XRT. We added a size bar in panel **c** and provided WT BMDM data as a control to Sirp $\alpha^{-/-}$ BMDM according reviewer's suggestion.

10. Throughout numbers of mice- n- in each group should be given. They are in some cases, but not in others Fig 3f for example.

Response: We thank the reviewer for pointing this out and have noted the numbers of mice in all figures.

11. No animal ethics information is provided.

Response: We apologize for missing this information. The animal ethics information has been provided in the revision (Page 22, marked in red).

12. Discussion indicates that "this study unveils an important mechanism by which malignant tumors resist RT..." This is not really true. What it reveals is that in the absence of Sirpa that immune response is enhanced. This is an aberrant situation and does not provide evidence for this general statement." They also say that "SIRP α deficient intratumoral macrophages respond to RT by manifesting a superior anti-tumor capacity. " They do not show that radiation has any effect on the macrophages or that they even need to be in the TME to have this effect.

Response: We agree with the reviewer that certain statements are not accurate. Indeed, the key finding of our study is that SIRP α -deficiency enhances the anti-tumor immune response. We have revised the statements accordingly (Page 18, marked in red). In the present study, local radiation is used as a tool to activate SIRP α -deficient macrophages. As we have shown in our previous study (Bian et al, *PNAS*, 2016, Ref #11), macrophages (even in the absence of the CD47-SIRP α signaling axis) require activation to phagocytize CD47-expressing self-cells, such as cancer cells. Local radiation causes cancer cell damage and upregulation/release of DAMPs such as HMGB1, DNA and calreticulin, etc. which in turn, activate macrophage phagocytic function. As shown in new Extended Data, Fig. 1, SIRP α deficient macrophages phagocytized irradiated (8Gy) but not non-irradiated MC38 cells. Therefore, tumor-localized irradiation is necessary for initiating the anti-tumor activity of SIRP α -deficient macrophages. Without IR, there was no tumor suppression/elimination in SIRP α -deficient mice. Our studies have also shown that other inflammatory cytokines or TLR agonists, such as IL-1 β , IL-6 and LPS, can initiate the phagocytosis of cancer cells by SIRP α -deficient macrophages (new Extended Data, Fig. 1a).

13. Extended Figure 9 introduces two new cell lines and a completely different experiment. It should be removed or else requires more thorough exploration.

Response: We removed extended Figure 9 from the revision according to the reviewer's comment.

References

Wang B, Li Q, Qin L, Zhao S, Wang J, Chen X. 2011. Transition of tumor-associated macrophages from MHC class II^{hi} to MHC class II^{low} mediates tumor progression in mice. *BMC Immunology* 12: 43.

Zhu D, Pan C, Li L, Bian Z, Lv Z, Shi L, Zhang J, Li D, Gu H, Zhang CY, Liu Y, Zen K. 2013. MicroRNA-17/20a/106a modulate macrophage inflammatory responses through targeting signal-regulatory protein α . *J Allergy Clin Immunol* 132: 426-36.e8.

Kong XN, Yan HX, Chen L, Dong LW, Yang W, Liu Q, Yu LX, Huang DD, Liu SQ, Liu H, Wu MC, Wang HY. 2007. LPS-induced down-regulation of signal regulatory protein α contributes to innate immune activation in macrophages. *J Exp Med* 204: 2719-31.

Reviewer #3

The aim of this study was to explain how SIRP α ^{-/-} status had improved RT efficacy on three preclinical cancer models. The authors demonstrated that SIRP, expressed by macrophages, regulated antigen presentation and tumor specific cytotoxic T cells radio-induced activation. This is a very interesting work, especially in this current era whereby radio-induced immune response is currently described to be depend on APC activation followed by CD8⁺ activation tumor antigen specific. This study highlight an important role of macrophages in this mechanism. Authors detailed each step of experiments to facilitate the understanding of their objectives for uninitiated readers. The conclusions and perspectives of this study will allow to develop new therapeutic to improve radiation therapy efficacy. Nonetheless while interesting, there are some major and minor limitations of the current data.

1. The Figure 1 is clear. To improve it I suggest adding the RT dose used in the Fig. 1b. The author have evaluated effect of different RT doses delivered using one or several fractions. I'm not sure that the Fig. 1e presents a real interest here. However it's unexpected that RT present no effect for WT model whatever the dose! Concerning MC38 and KPC, I invite the authors to insert in their references Jones et al., *EMBO Molecular Medicine* 2018 (10:e9342) and to discuss these results please! Moreover, Jones et al. have also observed a tumor growth delay after 1x10Gy which was improved after RT + anti-PDL1.

Response: First, we greatly appreciate the reviewer's positive evaluation of our work. According to the reviewer's suggestion, we have extensively re-examined the effect of RT on MC38 and KPC tumor growth in both Sirp α ^{-/-} and WT mice using different doses of radiation and selecting different sizes of tumors. We employed three IR dosages (4, 8 and 15Gy) and separated tumors into three groups based on the tumor volume: small (<100 mm³), medium (100-400 mm³) and large (400-600 mm³) (new Fig. 1b). In agreement with the study by Jones et al, we found that, for small tumors in WT mice, RT alone displayed dose-dependent inhibition of tumor growth (new Fig. 1d). However, RT failed to control the growth of larger tumors (>200 mm³) in WT mice. In contrast, a single fraction of IR in Sirp α ^{-/-} mice completely eliminated small tumors and dose-dependently suppressed the growth of medium or large tumors (new Fig. 1d). To better illustrate differences in tumor growth, additional statistical analyses including Kaplan Meier survival curves and Mantel-Cox Log-Rank Tests have been provided for all tumor-bearing mice. We have revised the text based on these results and inserted the study by Jones et al. (*EMBO Molecular Medicine* 2018, new Ref#21) into the reference accordingly (Page 6-7, revisions are in red).

Concerning the Suppl Fig 1, the authors used two dose rates to deliver 20Gy. For SIRP α ^{-/-}, they highlighted a better secretion of IL6, TNF α , IL12 and IL1 β after 20Gy delivered using 2Gy/min to compare to 1.2Gy/min. It's an interesting result which needs to be discuss!

Response: Indeed, we had explored different dose rates of radiation in our study. We found that a higher dose rate of IR (2Gy/min) had a better effect on tumor regression and proinflammatory cytokine induction in Sirp α ^{-/-} mice than a lower dose rate (1.2Gy/min). However, a higher dose rate of IR could exaggerate the inflammatory response in Sirp α ^{-/-} mice, which sometimes compromised mouse survival. Although the underlying mechanism remains unknown, it may be related to more tumor cell damage or

death caused by a higher dose rate of IR. As a higher dose rate of IR causes more tumor damage or death than a lower dose rate of IR (Lohse et.al. *Radiotherapy and Oncology*,101, Issue 1, October 2011; Turner et al. *Radiat Res.* 2015 Mar; 183(3)), more damage-associated molecular pattern (DAMP) molecules would be produced by tumor cells following a higher dose rate of IR compared to lower dose rate of IR. Given that Sirp $\alpha^{-/-}$ macrophages (not WT macrophages) are highly sensitive to activation by DAMPs and subsequently phagocytize cancer cells (Extended Data, Fig. 1), more tumor cell damage or death precipitated by a higher dose rate of IR may accelerate both phagocytosis of tumor cells and cytokine production by Sirp $\alpha^{-/-}$ macrophages.

2. The figure 2 investigated the radio-induced abscopal effect for SIRPa $^{-/-}$ model. Interestingly author demonstrated this effect using a standard contralateral tumor injection and also using un-irradiated intra peritoneal lesion. In the Fig. 2b experiments, authors observed an effect of initial tumor volume on response. As previously this result was not explained and discussed in the manuscript. It could be interesting to compare TME for tumor <100mm³ and >200mm³ which could provide some answers.

Response: We appreciate the reviewer's insight on this matter. Indeed, we found that the tumor volume at the time of primary tumor irradiation affected the extent to which the abscopal response occurred. According to the reviewer's suggestion, we compared the TME in tumors whose volume was <100mm³ or >200mm³. As shown in new Extended Data Fig. 11a, there was significantly more infiltration of CD8 Tc into small non-irradiated tumors than large non-irradiated tumors, supporting the role of Tc in abscopal remission of small tumors. In contrast, the TME of large non-irradiated tumors was still dominated by pro-tumor myeloid cells. Moreover, as the abscopal effect is dependent upon activation of anti-tumor Tc, we have moved the abscopal effect results (original Fig. 2) to Fig. 7, following the analysis of Tc activation by tumor antigen-presenting Sirp $\alpha^{-/-}$ macrophages.

3. The figures 3 clearly demonstrated that the RT efficacy for SIRPa $^{-/-}$ model was dependent on macrophages. All experiments were well explained except the last graph of the Fig. 3d. The reader will need some explanations in the legend for this experiment.

Response: Intratumoral macrophages generally play a pro-tumor role and serve as a major mechanism underlying tumor resistance to RT. The study by Jones et al. (*EMBO Molecular Medicine* 2018, 10:e9342, new Ref#21) clearly showed that simply depleting intratumoral macrophages could enhance the efficacy of RT anti-tumor therapy. Our study here finds SIRP α as a master controller of macrophage-mediated immune responses. In other words, depletion of SIRP α can turn macrophages from pro-tumor, immune suppressive cells into anti-tumor and immune active phagocytes. According to the reviewer's comment, we have provided more information to explain the experiment presented in the original Fig. 3d (new Fig. 2e) (Page 8, the changed parts are marked in red), as well as in the figure legend of new extended Fig. 2a.

4. The figure 4 demonstrated that the SIRPa $^{+/+}$ model lacked the capacity for immunogenic antigen presentation. To improve the understanding of the fig. 4a author could insert CD45 in x axis legend. In the sup fig 6 authors details their gating process to characterize each immune cells. Please explain how

you can highlight a radio-induced decrease in macrophage in Fig. 4b whereas they don't observe significant modification of myeloid cell quantity (which include macrophages!) in figure 4c (for SIRP α -/- model). Could you please explain and discuss these results?

Response: According to the reviewer's suggestion, we have inserted CD45 in the x axis legend of new Fig. 3a. Indeed, original Fig. 4b (now Fig. 3a) showed that the percentage of intratumoral macrophages in total CD45⁺ leukocytes in SIRP α ^{-/-} mice decreased following IR compared to that in WT mice. However, myeloid subtype analysis (new Fig. 3b) indicated that the number of inflammatory PMN in SIRP α ^{-/-} TME significantly increased, which likely compensated for the decreased number of macrophages and led to no significant change in the total quantity of myeloid cells. Based on the new results (new Fig. 3b), we revised the manuscript accordingly (Page 9-10, marked in red).

5. Finally, authors have to investigate if the phagocytosis capacity of macrophages is modified by SIRP depletion. There are some explorations about this in supplemental data which need to be explain and discuss in the manuscript.

Response: We appreciate the reviewer's comment on this critical issue. Given that SIRP α depletion is critical for macrophages to phagocytize tumor cells, we have provided results derived from different assays showing the enhancement of macrophage phagocytosis capacity by SIRP α depletion in the revision. As shown in new Extended Data Fig. 1, in line with our previous finding that Sirp α ^{-/-} macrophages activated by inflammatory cytokines or TLR agonists can phagocytize CD47-expressing self-cells, treatments with LPS, IL-1 β , IL-6, as well as DAMP molecules CpG (tumor-DNA mimetic) and HMGB1 released by tumor cells under radiation (He et al. *Cell Death & Disease*, 2018, new Ref #16), all strongly promoted Sirp α ^{-/-} macrophages but not WT macrophages phagocytosis of MC38 cancer cells in vitro (Extended Data Fig. 1a). Sirp α ^{-/-} macrophages also phagocytized irradiated MC38 cells (8Gy) but not non-irradiated MC38 cells (Extended Data Fig. 1b), suggesting that irradiated MC38 express certain DAMPs that activate Sirp α ^{-/-} macrophage phagocytic function toward CD47-expressing cancer cells. Indeed, we detected an increase in the level of calreticulin (CRT) on the surface of irradiated MC38 cells (Extended Data Fig. 1c, left), suggesting that CRT may be one DAMP expressed on irradiated MC38 cells. As cell surface CD47 level and pattern did not seem to decrease on 8Gy-treated MC38 cells (Extended Data Fig. 1c, right), we think that those irradiated cancer cells are partially damaged but have yet to enter apoptosis. This may be important because normally clearance of apoptotic cells by phagocytes does not lead to immunogenic antigen presentation, rather it often induces tolerance. Additional studies using medium comprising soluble factors secreted by irradiated or non-irradiated MC38 cells (conditioned medium) also found that irradiated but not non-irradiated MC38 conditioned medium promoted the phagocytosis of MC38 cells by Sirp α ^{-/-} macrophages, suggesting that irradiated MC38 cells may release certain DAMPs that activate the phagocytic function of Sirp α ^{-/-} macrophages (Extended Data Fig. 1d). Supporting this, analysis of downstream signaling in macrophages incubated with irradiated cancer cells or their conditioned medium clearly showed that treatment with irradiated cancer cells or their conditioned medium enhanced the phosphorylation of P65 and P38, two downstream signaling molecules indicative of TLR pathway activation (Dorrington et al. *Front. Immunol.*, 2019, new Ref #14), in Sirp α ^{-/-} macrophages compared to WT macrophages (Extended

Data, Fig. 1e). Taken together, these results suggest that Sirp $\alpha^{-/-}$ macrophages but not WT macrophages are more sensitive to activation by DAMPs released by irradiation-damaged (but not apoptotic yet) cancer cells.

References

- Lohse, I. et al. Effect of high dose per pulse flattening filter-free beams on cancer cell survival. *Radiotherapy and Oncology* 101, 226-232, doi:<https://doi.org/10.1016/j.radonc.2011.05.072> (2011).
- Turner, H. C. et al. Effect of dose rate on residual γ -H2AX levels and frequency of micronuclei in X-irradiated mouse lymphocytes. *Radiat Res* 183, 315-324, doi:10.1667/RR13860.1 (2015).

Reviewer #4

Bian and colleagues showed that SIRP α -deficiency elicited potent adaptive immune response following local radiation in murine syngeneic tumor xenograft models. After initially phenotypically demonstrating complete tumor xenograft rejection in SIRP α ^{-/-} deficient mice receiving local radiation treatment, the authors investigated the establishment of anti-tumor adaptive immune responses and pro-inflammatory remodeling of the tumor microenvironment. The authors suggested the potent anti-tumor adaptive immune responses were orchestrated via antigen-capture by SIRP α ^{-/-} macrophages and subsequently cross-priming and activated CTLs. Most experiments were well designed and executed with necessary controls included.

Major points:

1) Very interestingly, the authors suggested CD47 blockaded by intratumoral delivery of either functional antibody or soluble murine SIRP α ecto domain fell short to recapitulate the results observed in SIRP α ^{-/-} mice. However, this conclusion would be further strengthened using CD47 KO tumor xenograft inoculated in WT mice in addition to pharmacological blockade as shown in this manuscript, to compare whether CD47 KO tumor xenograft inoculated in WT mice achieves the same tumor rejection effect as observed in WT tumor xenograft in SIRP α ^{-/-} mice.

Response: We greatly appreciate reviewer's the positive evaluation of our work. Indeed, as CD47 blockade by either functional antibody or soluble SIRP α ectodomain fails to recapitulate the results observed in SIRP α ^{-/-} mice, we argue that there is a CD47-independent pathway by which activated SIRP α ^{-/-} macrophages exert anti-tumor responses. This notion is also supported by our *in vitro* phagocytosis results. As shown in new Extended Data, Fig. 1, anti-CD47 Ab or SIRP α ectodomain only moderately enhanced phagocytosis of MC38 cells by WT macrophages activated by various stimuli (new Extended Data, Fig. 1a) or phagocytosis of irradiated MC38 cells by WT macrophages (new Extended Data, Fig. 1b). In both cases, phagocytosis of MC38 cells was significantly less than that by SIRP α ^{-/-} macrophages under the same condition.

According to reviewer's suggestion, we obtained CD47-null MC38 cells from Dr. Fu (University of Texas Southwestern Medical Center) and engrafted CD47-KO MC38 in WT mice to determine whether CD47-KO tumors inoculated in WT mice achieves the same tumor rejection effect as observed in WT tumors in SIRP α ^{-/-} mice. As shown in the figure below, while both cell lines had a similar growth rate *in vitro* (panel **a**), CD47 KO MC38 cells grew significantly slower than WT MC38 cells especially at a later stage in WT mice (panel **b**, left). The slower growth of this CD47-KO MC38 cell line in WT mice is in agreement with a previous report (Liu et al. *Cell Reports* 24, Issue 8, 2018, 2101-2111). When CD47-KO MC38 tumors were established (~200mm³), local radiation (8Gy) was applied. As shown in panel **b** (middle), CD47-KO MC38 tumors regressed and finally were eliminated following IR, although this tumor remission proceeded at a different rate than WT MC38 in SIRP α ^{-/-} mice (red dot line). In line with this, TME analysis also showed Tc infiltration in CD47-KO MC38 tumor following local IR, but this occurred a few day later than Tc infiltration into WT MC38 tumors in SIRP α ^{-/-} mice (panel **b**, right). Given that CD47 is a SIRP α ligand and ligation of SIRP α by CD47 activates inhibitory SIRP α signaling, these results are in line with our conclusion that SIRP α serves as master regulator controlling

macrophage anti-tumor immunity. It may also suggest that CD47 blockade by functional antibody or soluble SIRP α ectodomain is not able to completely abrogate CD47-SIRP α signaling in the TME, as that achieved by the CD47-KO MC38 cells. However, given that there is stunted growth of CD47-KO MC38 tumors compared to parental MC38 tumors in WT mice, there is likely a different mechanism underlying its regression in WT mice that had received RT. Interestingly, we also observed a significant defective angiogenesis in CD47-KO MC38 tumor tissue compared to WT MC38 tumor in WT mice following local RT (panel **c-d**). Although the mechanism underlying deficient angiogenesis in CD47-KO MC38 tumors remains unknown and requires further study, this may provide a different explanation for the observed rejection effect against this CD47-KO MC38 tumor in WT mice.

Reviewer only Figure 2. a) In vitro growth of MC38 and CD47KO MC38 cells. b) *In vivo* growth and TME analysis of MC38 and CD47KO MC38 tumor with or without irradiation. c) CD31 staining of endothelium cells in CD47KO and control MC38 tumor with and without irradiation. d) representative images angiogenesis in MC38 and CD47KO MC38 tumor with and without irradiation.

2) Data representation is a major obstacle hindering readers to analysis and comprehend. While the authors tried to present data in a more comprehensive way, due to the limited space of a figure yet too many panels presented, it's challenging to read in detail for almost all figures. The conclusion was strengthened after validated in multiples models, as the authors did with MC38, KPC and Pan02 lines. However, KPC and Pan02 models provided minor additional information that was absent from MC38 model. An alternative way would be, for example, to move figures related to KPC and Pan02 into supplementary information while presenting only MC38 related data in the main Figures.

Response: We apologize for the difficulty to go through many panels in each figure. According to the reviewer's comment, we moved some KPC and Pan02 data into the Extended Data section (new Extended Data Fig. 5 and Fig. 10, etc.) and focused on MC38. We also provided more schematic figures, including new Fig. 1b, and new Extended Data Fig. 9a, etc. for illustrating experiment design.

3) In the “macrophages present antigens to activate tumor-specific Tc in situ” section, the conclusion of the first paragraph “These results suggest that ICD and DAMPs-activated Sirp $\alpha^{-/-}$ macrophages drive Tc expansion in situ independent of peripheral lymphoid organs” was insufficiently supported based on the results demonstrated in fig. 7a. In this experiment, the authors failed to demonstrate neither tumor cell apoptosis nor the release of DAMPs and specifically which DAMPs were released and thus macrophages sensed and activated.

Response: We appreciate the reviewer’s constructive suggestion on this. Indeed, activation of SIRP $\alpha^{-/-}$ macrophages to phagocytize tumor cells is essential for SIRP $\alpha^{-/-}$ macrophage-based anti-tumor therapy in the present study. We thus provided results demonstrating phagocytosis of tumor cells by SIRP $\alpha^{-/-}$ macrophages activated by various stimuli especially the DAMPs released by irradiated tumor cells. As shown in new Extended Data Fig. 1, in line with our previous finding that Sirp $\alpha^{-/-}$ macrophages activated by inflammatory cytokines or TLR agonists can phagocytize CD47-expressing self-cells, treatments with LPS, IL-1 β , IL-6, as well as DAMP molecules CpG (tumor-DNA mimetic) and HMGB1 released by tumor cells under radiation (He et al. *Cell Death & Disease*, 2018, new Ref #16), all strongly promoted Sirp $\alpha^{-/-}$ macrophages but not WT macrophages phagocytosis of MC38 cancer cells in vitro (Extended Data Fig. 1a). Sirp $\alpha^{-/-}$ macrophages also phagocytized irradiated MC38 cells (8Gy) but not non-irradiated MC38 cells (Extended Data Fig. 1b), suggesting that irradiated MC38 express certain DAMPs that activate Sirp $\alpha^{-/-}$ macrophage phagocytic function toward CD47-expressing cancer cells. Indeed, we detected an increase in the level of calreticulin (CRT) on the surface of irradiated MC38 cells (Extended Data Fig. 1c, left), suggesting that CRT may be one DAMP expressed on irradiated MC38 cells. As cell surface CD47 level and pattern did not seem to decrease on 8Gy-treated MC38 cells (Extended Data Fig. 1c, right), we think that those irradiated cancer cells are partially damaged but have yet to enter apoptosis. This may be important because normally clearance of apoptotic cells by phagocytes does not lead to immunogenic antigen presentation, rather it often induces tolerance. Additional studies using medium comprising soluble factors secreted by irradiated or non-irradiated MC38 cells (conditioned medium) also found that irradiated but not non-irradiated MC38 conditioned medium promoted the phagocytosis of MC38 cells by Sirp $\alpha^{-/-}$ macrophages, suggesting that irradiated MC38 cells may release certain DAMPs that activate the phagocytic function of Sirp $\alpha^{-/-}$ macrophages (Extended Data Fig. 1d). Supporting this, analysis of downstream signaling in macrophages incubated with irradiated cancer cells or their conditioned medium clearly showed that treatment with irradiated cancer cells or their conditioned medium enhanced the phosphorylation of P65 and P38, two downstream signaling molecules indicative of TLR pathway activation (Dorrington et al. *Front. Immunol.*, 2019, new Ref #14), in Sirp $\alpha^{-/-}$ macrophages compared to WT macrophages (Extended Data, Fig. 1e). Taken together, these results suggest that Sirp $\alpha^{-/-}$ macrophages but not WT macrophages are more sensitive to activation by ICD/DAMPs released by irradiation-damaged (but not apoptotic yet) cancer cells.

4) Supplementary Fig. 6, gating strategy. The authors failed to gate singlet prior to analysis. Without singlet gating, all downstream analyses were rendered problematic, especially for xenograft tumor tissue

as they are notoriously sticky even after thorough digestion and passed through strainer. Failure to exclude doublets could introduce false positive in flow cytometry analysis. The authors should re-analyze flow cytometry data after making sure singlet gating is applied.

Response: We thank the reviewer for pointing out this critical issue. Indeed, xenograft tumor tissue are sticky even after thorough digestion and passed through strainer. In the present study, singlet gating was performed prior to each analysis. We added this into our gating strategy (new Supplementary Fig. 7).

Minor points:

Text related

1) Sentence “Despite the absence of the TDLN, irradiated tumor explants from *Sirpα*^{-/-} mice exhibited robust expansion of Tc similar to that which occurred *in vivo*.” Please add indication to which experiment in figures or citation to which study did the “that which occurred *in vivo*” refers to.

Response: We apologize for the confusion caused by this sentence. This sentence refers to Fig.5b (new Fig.4a), which demonstrated that irradiation of MC38 tumors in *Sirpα*^{-/-} mice led to a rapid expansion of intratumoral Tc. We have fixed it accordingly (Page 14, marked in red). Indeed, in new Fig. 6, we performed an *ex vivo* assay to demonstrate that activated intratumoral *Sirpα*^{-/-} macrophages by local tumor radiation are capable of inducing proliferation and activation of tumor-specific Tc. In the experiment, tumors without tumor-draining lymph nodes (TDLN) were excised following IR, and then dissociated into individual cells and cultured *ex vivo* (Fig. 6a). Successful amplification of tumor-specific CD8 Tc not CD4 Th from the tumor dissociates, which include both tumor antigen-presenting *Sirpα*^{-/-} macrophages and TIL, suggests that irradiated tumor cell DAMPs activated *Sirpα*^{-/-} macrophages that then drive Tc expansion independent of peripheral lymphoid organs. Additionally, we also used *Sirpα*^{-/-} BMDMs to phagocytose and process tumor antigens and induced expansion of tumor-specific Tc from isolated TILs. Adoptive transfer of these Tc into tumor-bearing WT mice confirmed their anti-tumor function (Fig. 6i-j). The *ex vivo* amplification of Tc present within the dissected TDLN-free tumor tissue suggests that robust expansion of Tc largely occurred *in situ* in irradiated tumor-bearing *Sirpα*^{-/-} mice.

2) Please provided detailed experiment method description regarding the *ex vivo* culture of tumor tissue experiment in fig. 7a.

Response: We have provided a detailed description of the method used for *ex vivo* culture of tumor tissue in new Fig. 6 (Page 30, marked in red).

3) Data transparency. In Figure legend of Figure 1. Please indicate exactly how many experiments were repeated, instead of “at least three independent experiments”

Response: We have indicated the exact number of experiments and number of mice in Figure 1.

Figure related

- Fig. 4a, annotations of X- and Y- axis are missing.
- Fig. 4c, statistic P value missing for WT panel.
- Fig. 4h, authors should demonstrate transcriptome profiling heatmap of each biological replicates separately (n=2-4 as described in figure legend) instead of the representative sample or group average as shown.

Response: We appreciate the reviewer for pointing out these errors and have fixed them accordingly.

References

Liu X, Liu L, Ren Z, Yang K, Xu H, Luan Y, Fu K, Guo J, Peng H, Zhu M, Fu Y-X. 2018. Dual Targeting of Innate and Adaptive Checkpoints on Tumor Cells Limits Immune Evasion. *Cell Reports* 24: 2101-11

REVIEWERS' COMMENTS

Reviewer #2: report not provided

Reviewer #3 (Remarks to the Author):

Dear Authors

thank you for the changes made in this manuscript. You have taken into account my remarks and have answered clearly to the different ones.

I sincerely think that this new version is more convincing and rigorous than the previous one.

For my part, the answers of the authors responding to all of my comments, I am in favor of the publication of this version.

Reviewer #4 (Remarks to the Author):

The authors have adequately addressed my concerns. Thank you!

Reviewer #3:

Dear Authors

thank you for the changes made in this manuscript. You have taken into account my remarks and have answered clearly to the different ones.

I sincerely think that this new version is more convincing and rigorous than the previous one.

For my part, the answers of the authors responding to all of my comments, I am in favor of the publication of this version.

Response: Thank you for your kind words and encouragement.

Reviewer #4:

The authors have adequately addressed my concerns. Thank you!

Response: Thank you for your thorough review of our resubmission and positive comments.